# Cryo-EM Structure of the relaxosome, a complex essential for bacterial mating and the spread of antibiotic resistance genes

Sunanda M. Williams [1]✉, Sandra Raffl[2], Sabine Kienesberger [2], Aravindan Ilangovan [1,3], Ellen L. Zechner [2] & Gabriel Waksman [1,4]✉

Bacterial mating, or conjugation, was discovered nearly 80 years ago as a process transferring genes from one bacterial cell (the donor) to another (the recipient). It requires three key multiprotein complexes in the donor cell: a DNA-processing machinery called the relaxosome, a double-membrane spanning type 4 secretion system (T4SS), and an extracellular appendage termed pilus. While the near-atomic resolution structures of the T4SS and pilus are already known, that of the relaxosome has not been reported to date. Here, we describe the cryo-EM structure of the fully assembled relaxosome encoded by the paradigm F plasmid in two different states corresponding to distinct functional steps along the DNA processing reaction. By varying the structures of model DNAs we delineate conformational changes required to initiate conjugation. Mutational studies of the various protein-protein and protein-DNA interaction hubs suggest a complex sensitive to trigger signals, that could arise from cell-to-cell contacts with recipient cells.

The nucleoprotein complex termed relaxosome contains the relaxase and other auxiliary transfer proteins, which assembles at the "origin of transfer" (*oriT*) region of a plasmid and prepares it for transfer to another bacterial cell[1,2]. This process whereby plasmids spread among bacterial populations through cell-to-cell contact is termed bacterial conjugation, an important way to exchange genetic material and plays a crucial role in bacterial adaptation and evolution[3]. It is also one of the most effective means by which antibiotic resistance genes spread among bacterial populations, resulting in antimicrobial resistance (AMR)[4–8]. AMR is considered one of the most severe global public health and development threats of this century[9]. The large family of F plasmids which we are focusing on here, are the main carriers of antibiotic resistance genes in *E. coli* and historically associated with resistance or 'R factors'[10].

All DNAs transferred via conjugation do so from an essential sequence called *oriT*, a stretch of DNA about 400 base pairs (bp) in the F plasmid[11–13]. The F plasmid relaxosome complex contains, in addition to *oriT*, three plasmid-encoded proteins including the DNA-processing enzyme TraI (generally known as the "relaxase") and 2 accessory proteins TraY and TraM, and IHF, a protein encoded by the bacterial genome (Supplementary Fig. 1a). To date, multiple binding sites for these proteins have been characterised on *oriT* (2 TraI binding sites (*traI*$_{helicase}$ and *traI*$_{TE}$), 2 IHF binding sites (*IHFa,b*), 3 TraM binding sites (*sbmA,B,C*) and 2 TraY binding sites (*sbyA,C*); details in Supplementary Fig. 1b).

TraI is a multi-domain protein that consists of an N-terminal trans-esterase (TE) domain that contains its "relaxase" activity, followed by two helicase domains (one vestigial (VH), the other active (AH)), and a C-terminal domain (CTD) responsible for interactions with TraM (Supplementary Fig. 1c, d)[14–18]. IHF, TraY, and TraM are non-catalytic proteins. IHF, a binary complex of two very similar proteins, IHFα and IHFβ, is known to induce a sharp U-turn bend on its binding site (Supplementary Fig. 1e)[19]. TraM is responsible for binding to both its cognate *oriT* regions and to the C-terminal tail of TraD (Supplementary

[1]Institute of Structural and Molecular Biology, School of Natural Sciences, Birkbeck College, Malet Street, London WC1E 7HX, UK. [2]Institute of Molecular Biosciences, University of Graz, BioTechMed-Graz, Humboldtstrasse 50, 8010 Graz, Austria. [3]Centre for Molecular Cell Biology, School of Biological and Behavioural Sciences, Queen Mary University of London, Newark Street, London E1 2AT, UK. [4]Institute of Structural and Molecular Biology, Division of Biosciences, Gower Street, University College London, London WC1E 6BT, UK. ✉e-mail: sunanda.williams@bbk.ac.uk; g.waksman@bbk.ac.uk

Fig. 1f)[20,21], a T4SS protein that functions as recruitment platform for the relaxosome. TraY has DNA-bending ability and is known to stimulate TraI activity when bound to *oriT*[22].

Once formed, the relaxosome is recruited to the T4SS[23] and the relaxosome-T4SS super-complex may rest quiescent until contacts between donor and recipient cells are established through the pilus[24], activating the relaxosome. Activation initiates a yet uncharacterised cascade of molecular events resulting in the nicking of one of the plasmid DNA strands (the T-strand to be transferred) at a site called "*nic*" (Supplementary Fig. 1b) by the TE activity of TraI. The TraI catalysed nicking is stimulated by the other proteins in the relaxosome[22,25]. Nicking is followed by covalent attachment of TraI to the free 5′ phosphate resulting from the nicking reaction yielding a protein-ssDNA conjugate that, once unwound from its complementary strand (the R strand), is transported through the T4SS into the recipient cell[26–29]. To achieve this, another molecule of TraI functions as a helicase, unwinding the dsDNA in the 5′ to 3′ direction[14,30,31]. Once transfer is completed, both the transferred T-strand and the copy retained by the donor (the R-strand) undergo replication to duplex DNA thereby increasing the abundance of the transferred genome (and any antibiotic resistance genes it may carry) within a bacterial population[32].

A fully functional assembly of the relaxosome is crucial for effective nicking and transfer of the plasmid. Mutation of a catalytic tyrosine in the TraI active site reduces plasmid transfer by ~ 5000-fold[33]. *oriT* was shown to have functional domains which individually contribute to nicking and transfer functions[34]. Deletions of *oriT* binding sites and/or in the absence of other relaxosome proteins, TraI can only bind to ssDNA and cannot initiate transfer from ds *oriT*[31].

Molecular events controlling the start of conjugative gene transfer are yet to be understood. Some structural information is available on individual protein components of the relaxosome in gram-negative (Supplementary Fig. 1d–f)[14,19,35] and more recently in gram-positive bacteria[36–39]. However, crucially, there is to date no information on the structure of a fully or even partially assembled relaxosome.

## Results and Discussion

### Assembly of the relaxosome and identification of DNase I footprints of protein components

We first confirmed the feasibility of purifying a fully-assembled relaxosome from purified components. The four proteins were mixed in molar excess of a purified DNA of 316 bp (*oriT*$_{316}$) encompassing all known binding sites for each protein and the resulting complex (*oriT*$_{316}$-R) was purified (Fig. 1a, b).

To map the footprint of the fully assembled relaxosome complex, the purified *oriT*$_{316}$-R complex was subjected to DNase I cleavage to hydrolyse regions of *oriT* that are not protected upon complex formation. We show here (Fig. 1b) that most protected footprint sequences map to regions of *oriT*$_{316}$ extending from the *nic* site to include *traI*$_{TE}$, *IHFa*, *sbyC*, *sbyA* and partial *sbmC* sites. A single outlier was identified encompassing *IHFb* and *sbmB* sites.

From these results, we derived a dsDNA of 170 bp termed ds$_{-27_{+143}}$ in Figs. 2a, 1c and Supplementary Fig. 2 (the nomenclature of ss, ds, and ss/ds DNAs is explained in Fig. 1c). This DNA encompasses the protected region and additionally the *traI*$_{helicase}$ site and the full *sbmC* site. Using ds$_{-27_{+143}}$, a relaxosome complex termed "ds$_{-27_{+143}}$-R" was reconstituted from purified components and stabilised by crosslinking.

### Determination of the Cryo-EM structure of ds$_{-27_{+143}}$-R

The structure of the ds$_{-27_{+143}}$-R complex was derived from a "locally-refined" cryo-EM map with average resolution of 3.78 Å (Supplementary Figs. 3, 4a, g, 5a–d; Supplementary Table 1a, b; also see definition of "locally-refined" versus "global" in Supplementary Fig. 4). Cryo-EM is the most suitable structural biology method to use in this case as the structure is modular and cannot be crystallised. The density was of sufficient quality to build a model that includes (Fig. 2b): i- a large

region of dsDNA from base pairs +12 to +94 (sequence in bold in Fig. 2a), ii- the IHF α and β chains, iii- a train of three TraY molecules and iv- two sub-domains of the VH domain of TraI, termed 2 A, and 2B/2B-like (labelled VH$_{2A+2B/2B-like}$ in Fig. 2b; see details of the TraI sub-domain structure in Supplementary Fig. 1c, d)[17].

Overall, the salient feature of this structure is an asymmetric U-shaped dsDNA hairpin, generated by IHF binding. The region of ordered dsDNA in the density stretches over 83 bp, 52 bp on one side of IHF containing the TraY-binding sites (termed the TraY-binding arm of the dsDNA hairpin in Fig. 2b; bp$_{+43}$ to bp$_{+94}$) and 31 bp on the other side (termed the *nic* arm of the dsDNA hairpin in Fig. 2b; bp$_{+12}$ to bp$_{+42}$). On the former arm, three bound TraY proteins are observed, two repeated directly (TraY1 and TraY2) and the other inverted (TraY3). This train of three TraY molecules induces longitudinal bending of the dsDNA along its long axis, changing its trajectory. On the *nic* arm, a combination of sequence-induced dsDNA bending[40] between bp$_{+20}$ to bp$_{+34}$ and dsDNA bending induced by VH$_{2A+2B/2B-like}$ binding is observed (Supplementary Fig. 5f–h). Bends induced by IHF and accessory proteins may position the *nic* site for subsequent cleavage by TraI.

The VH$_{2A+2B/2B-like}$ module interacts with the two arms of the dsDNA hairpin and also with IHF and TraY. This dense network of interactions is seen in all relaxosome structures presented here (see details below).

### Towards reconstructing the entire relaxosome structure

We sought next to obtain a more complete relaxosome structure by experimenting with DNA sequences and ss/ds hybrid substrates. Previous X-ray crystallography of the TraI TE domain alone mapped its binding site to 11 nucleotides spanning *nic* (nucleotides -2 to +9 in Fig. 2c)[33,41]. Therefore, we designed a heteroduplex DNA termed "ss$_{-27_{+8}}$ds$_{+9_{+143}}$", containing a single-stranded (ss) T-strand region from −27 to +8 and a double-stranded (ds) region from +9 to +143 (Figs. 2c, 1c, Supplementary Fig. 2). The corresponding relaxosome complex is termed "ss$_{-27_{+8}}$ds$_{+9_{+143}}$-R". This structure was built into a locally-refined cryo-EM map with an overall resolution of 3.45 Å (Supplementary Figs. 4b, 6a; Supplementary Table 1a, b).

The following differences were readily apparent compared to the ds$_{-27_{+143}}$-R structure (Fig. 2d): i- the TE domain of TraI was entirely observable and could be built in the density as well as the portion of the T-strand bound to it (nucleotides -2 to +9), ii- dsDNA from base pairs +10 to +93, VH$_{2A+2B/2B-like}$, both chains of IHF, as well as the three TraY molecules present in the ds$_{-27_{+143}}$-R structure were better defined in the density and improved models could therefore be derived, and iii- density for the NTD and 1 A domains of VH (VH$_{NTD+1A}$) exhibited clear secondary structural elements in which a main-chain model could be fitted in density.

Additional densities were observed in the "global" density map of the ss$_{-27_{+8}}$ds$_{+9_{+143}}$-R complex (Supplementary Figs. 4b, h). These additional densities may correspond to the AH and CTD domains of TraI or/and to TraM, a protein that is also part of the complex. To locate these domains/proteins within these densities, we solved the cryo-EM structures of the ss$_{-27_{+8}}$ds$_{+9_{+143}}$-R complex where i- a version of TraI lacking the AH and CTD domains was used (TraI$_{1-863}$; complex termed ss$_{-27_{+8}}$ds$_{+9_{+143}}$-RΔ$_{AH+CTD}$; Supplementary Figs. 6b, 4c, i) or ii- TraM was omitted in complex formation (complex termed ss$_{-27_{+8}}$ds$_{+9_{+143}}$-RΔ$_{TraM}$ (Supplementary Figs. 6c, 4d, j). Results of global map superpositions for both ss$_{-27_{+8}}$ds$_{+9_{+143}}$-RΔ$_{AH+CTD}$ and ss$_{-27_{+8}}$ds$_{+9_{+143}}$-RΔ$_{TraM}$ with ss$_{-27_{+8}}$ds$_{+9_{+143}}$-R are shown in Fig. 3a, b, respectively. These identified the density for TraM and for TraI AH and CTD, the models[17,19,35] of which were subsequently fitted in the corresponding density. Further validation of TraM docking is detailed in Supplementary Fig. 9c. Together, these results provide the basis to reconstruct a fully-assembled relaxosome (Fig. 2e and Supplementary Data 1).

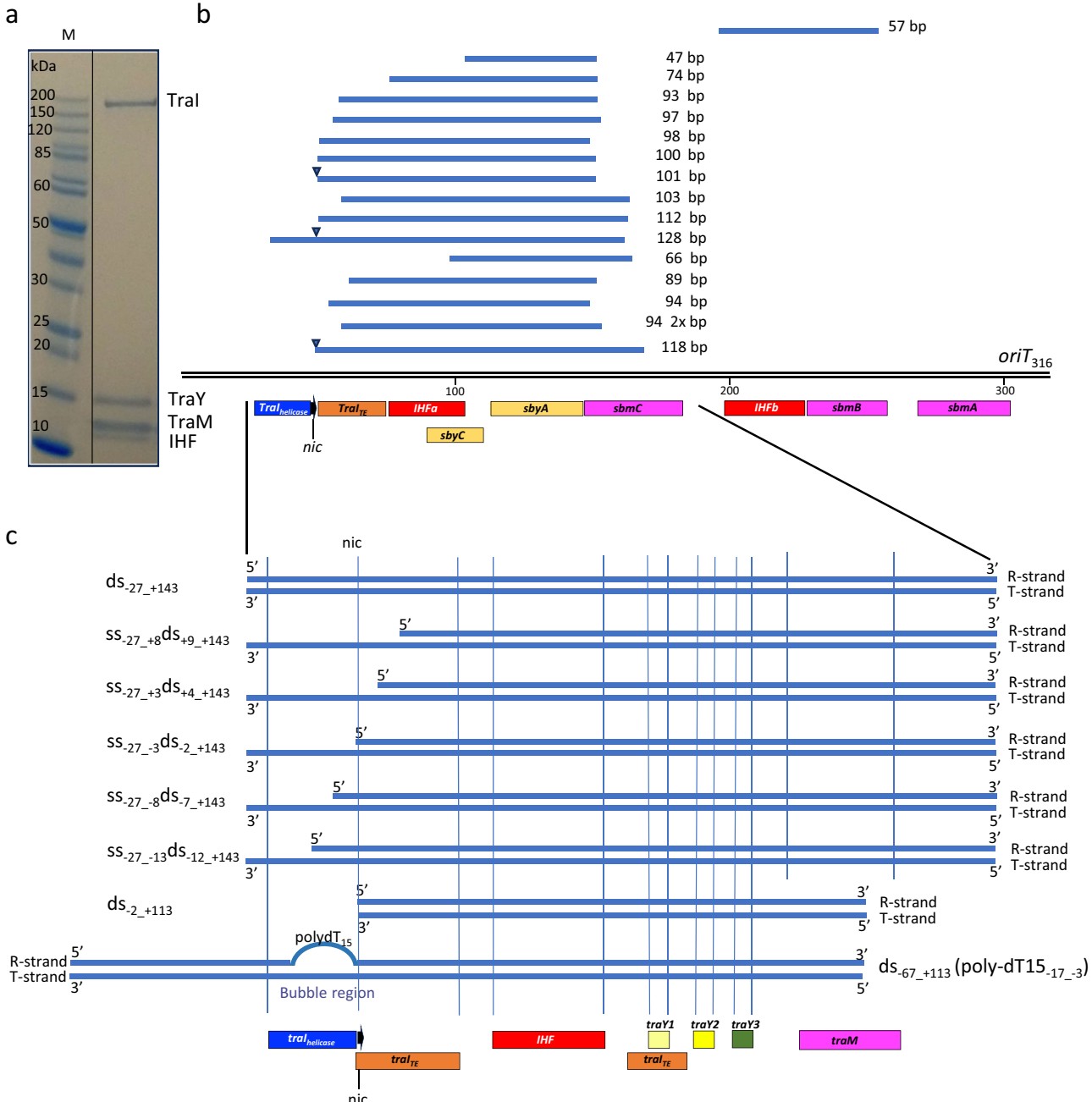

**Fig. 1 | Biochemical characterisation of the F plasmid relaxosome. a** SDS-PAGE gel of the relaxosome. Purified TraI, TraY, IHF, and TraM were mixed with *oriT₃₁₆* and unbound components were excluded by size-exclusion chromatography, leading to a purified relaxosome. Molecular Weight markers are labelled and *n* = 6 independent experiments. For the uncropped gel image see source data file. For structural studies, the complex was stabilised by cross-linking with glutaraldehyde. We do not believe that crosslinking introduces bias in our case, since our structure recapitulates prior biochemical and structural knowledge on individual proteins. **b** *oriT* sequences protected from DNase I nuclease cleavage. The sequenced fragments represented by blue lines are mapped against *oriT₃₁₆*. The length of each fragment is shown. The various binding sites for each relaxosome protein as defined by Ilangovan et al.[14] for TraI[14] and by Frost et al., and Lum et al.[11,69] for IHF, TraM, and TraY are shown under *oriT₃₁₆* using boxes colour-coded red for IHF, yellow for TraY, dark blue and orange for TraI_helicase and TraI_TE, and magenta for TraM. When present in the DNA fragment, the position of the *nic* site is shown by a black filled triangle. **c** DNAs used in this study. The DNAs are labelled according to the following notation: DNAs are numbered relative to the *nic* site on the T-strand. Therefore, *nic* serves as the origin, with the bases or base pairs 3′ to it being negatively numbered, and the bases and base pairs 5′ to it being positively numbered. This notation provides instant recognition of the position of *nic* in the DNA used. Furthermore, "ss" and "ds" DNA is used to indicate which region of *oriT* is single-stranded or double-stranded in the DNA used. For example, ss₋₂₇₊₈ds₊₉₊₁₄₃ is a DNA which is single-stranded from base +8 to −27 and double-stranded from +9 to +143. The various binding sites for each relaxosome proteins as discovered in this study are shown underneath. Box colour-coding is as in panel b except for the three TraY molecule train which are coloured pale yellow, bright yellow, and olive green for TraY1, TraY2, and TraY3, respectively. The position of the *nic* site (black arrow) is shown.

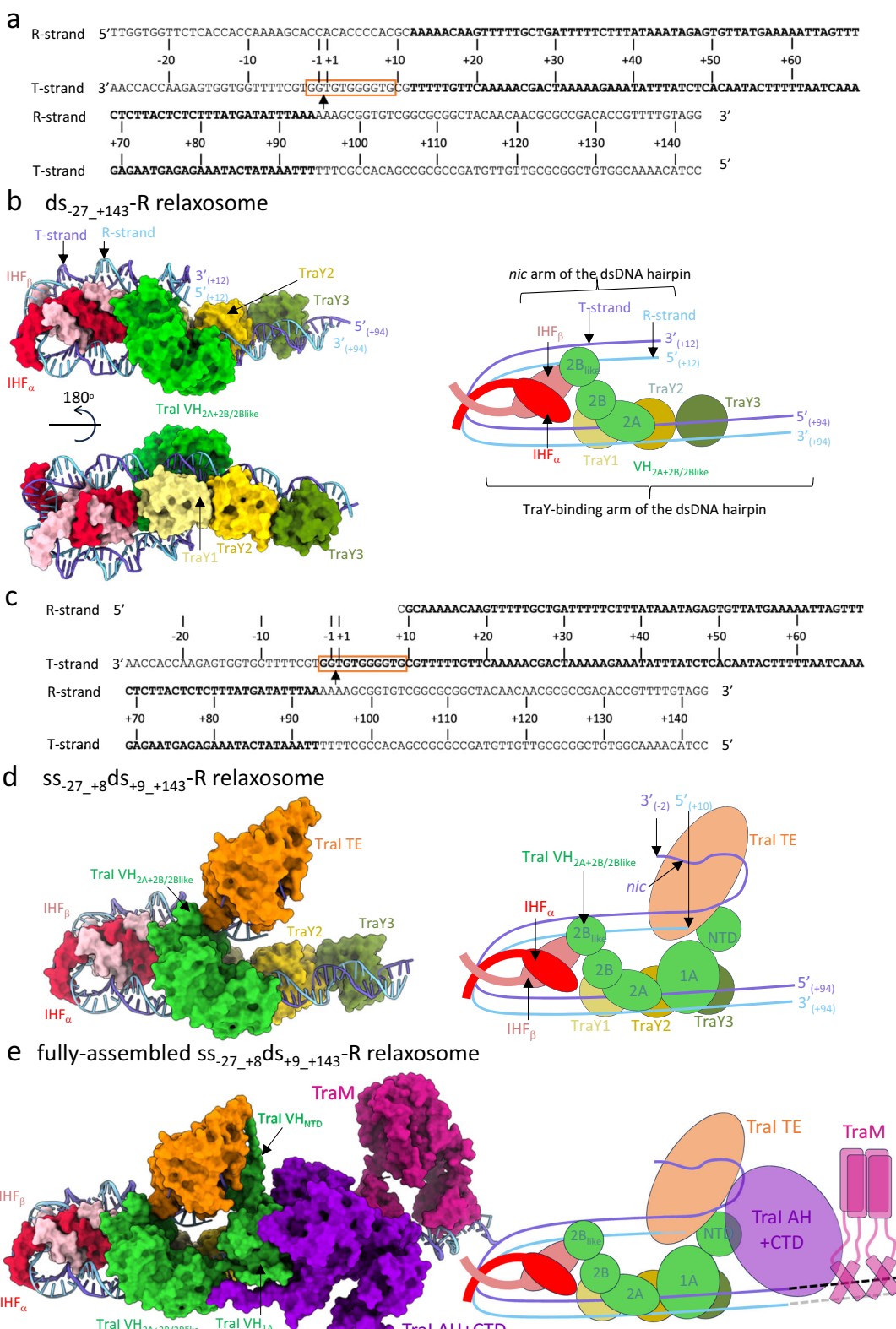

### Structure of the fully-assembled relaxosome

The structure of the ss$_{-27\_+8}$ds$_{+9\_+143}$-R complex consists of successive but overlapping interaction centres analogous to railway hubs organised along a rigid track of DNA (Fig. 4a). Four hubs each combining protein-DNA and protein-protein interactions can be defined. Hub 1 made of IHF/TraI VH$_{2A+2B/2B\text{-like}}$/TraY1 and dsDNA between base pairs +13 and +74 (Fig. 4b, c and Supplementary Fig. 7a, b), Hub 2 formed by a "train" of three TraYs and dsDNA between base pairs +61 and +90 (Fig. 5a and Supplementary Fig. 8), Hub 3 made of TraI TE, TraI VH$_{2B/2B\text{-like}}$, and part of the *nic* arm of the DNA hairpin that includes the ssDNA between -2 and +9 and dsDNA between +10 and +15 (Fig. 5b, c and Supplementary Fig. 9a, b), and Hub 4 made of TraM and dsDNA between base pair +93 and +117 (Supplementary Fig. 9c, d).

**Fig. 2 | Structures of the ds$_{-27\_+143}$-R relaxosome, the ss$_{-27\_+8}$ds$_{+9\_+143}$-R relaxosome, and the fully-assembled ss$_{-27\_+8}$ds$_{+9\_+143}$-R relaxosome. a** DNA used to solve the structure of ds$_{-27\_+143}$-R. Nomenclature and numbering is as in Fig. 1c. *nic* is indicated by an arrow. The region of DNA for which a model could be derived from the density is shown in bold. Replicative (R-strand) and transfer (T-strand) strands are indicated. Orange box locates the TE-binding site on the T-strand between bp$_{-2}$ and bp$_{+9}$. **b** Structure of ds$_{-27\_+143}$-R. This structure was derived from the "locally-refined" map shown in Supplementary Fig. 4a. Left: Two orientations of the complex. Proteins and DNA are shown in surface and ribbon, respectively, colour-coded dark and light blue for the T- and the R-strand respectively; red and pink for IHFα and IHFβ, respectively; pale yellow, bright yellow and olive green for TraY1, TraY2, and TraY3, respectively; and green for the only TraI part visible in the density for this complex, VH$_{2A+2B/2B-like}$. 5' and 3' end base pairs are labelled. Right: Schematic diagram of the structure in the orientation shown at upper left. The two arms of the dsDNA hairpin are labelled. Same colour-coding as at left is used. **c** DNA used to solve the structures of ss$_{-27\_+8}$ds$_{+9\_+143}$-R complex (Fig. 1c). *nic*, the region of DNA for which a model could be derived from the density, and the location of the TE-binding site on the T-strand are shown as in panel **a**. **d** Structure of ss$_{-27\_+8}$ds$_{+9\_+143}$-R. This model is derived from the locally-refined map described in Supplementary Fig. 4b. Proteins and DNA are shown in surface and ribbon, respectively, colour-coded as in panel **b**, except for the additional TE domain in orange. Left: Near-atomic resolution structure of the complex. Right: Schematic representation of the structure. 3' and 5' DNA ends are indicated as well as the location of the *nic* site. **e** Structure of the fully-assembled ss$_{-27\_+8}$ds$_{+9\_+143}$-R relaxosome. This model is derived from both the locally-refined and global maps reported in Supplementary Fig. 4b–d and h–j. Proteins and DNA are shown in surface and ribbon, respectively, colour-coded as in panel **d**. TraI AH + CTD is in violet and TraM is in magenta. Left: model of the complex. Right: Schematic diagram.

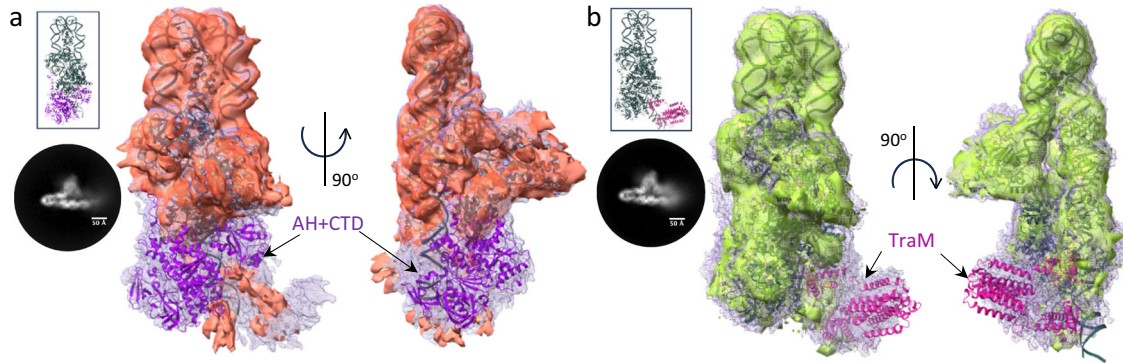

**Fig. 3 | Use of maps where parts of the relaxosome have been deleted to generate the complete structure of the relaxosome. a** Superposition of the unsharpened global maps of ss$_{-27\_+8}$ds$_{+9\_+143}$-R (in light blue mesh) and ss$_{-27\_+8}$ds$_{+9\_+143}$-RΔ$_{AH+CTD}$ (in light red surface). Maps details are reported in Supplementary Fig. 4h and i, respectively. A representative 2D class is shown on the left. The structure of ss$_{-27\_+8}$ds$_{+9\_+143}$-R is in dark grey ribbon. A model of TraI AH and CTD (in violet ribbon) is shown fitted in the additional density observed in ss$_{+8\_-27}$ds$_{+9\_+143}$-R, but not in ss$_{+8\_-27}$ds$_{+9\_+143}$-RΔ$_{AH+CTD}$. Two orientations, 90° apart, are shown at left (same orientation in inset) and right. The lack of resolution for TraI AH and CTD is not surprising. Firstly, TraI is a remarkably flexible protein. It unfolds readily to pass through the pilus during conjugative transfer and it is known to be unusually susceptible to mild-proteolytic cleavage either in its unbound form or bound to its TE ssDNA-binding site (*traI$_{TE}$* in Fig. 1b)[14,77]. It is known to be ordered only in its helicase-binding mode[14,78]. Moreover, both TraI AH and CTD together with TraM are located in a region of the DNA that is itself ill-defined, indicating flexibility of the DNA in this region. **b** Superposition of the unsharpened global maps of ss$_{-27\_+8}$ds$_{+9\_+143}$-R (in light blue mesh) and ss$_{-27\_+8}$ds$_{+9\_+143}$-RΔ$_{TraM}$ (in light green surface). Map details are reported in Supplementary Fig. 4h,j, respectively. A representative 2D class is shown on the left. The structure of ss$_{-27\_+8}$ds$_{+9\_+143}$-R is in dark grey ribbon. A model of TraM (in magenta ribbon) is shown docked in the additional density observed in ss$_{+8\_-27}$ds$_{+9\_+143}$-R but not in ss$_{+8\_-27}$ds$_{+9\_+143}$-RΔ$_{TraM}$. Two orientations, 90° apart, are shown at left (same orientation in inset) and right. To validate the TraM location, we also generated an extended model of the DNA that includes *sbmC* and observed that *sbmC* locates in the density region attributed to TraM. Moreover, the molecular interactions of TraM within this region of DNA are very similar to those identified previously by hydroxyl radical foot printing thereby providing another layer of confidence that the density for TraM has been properly ascribed (Fig. 5d and Supplementary Fig. 9c)[20].

Hub 1 is the largest hub in terms of buried surface area (~8000 Å$^2$). Interactions between IHF chains and between IHFαβ and its dsDNA-binding site (between bp+23 and bp+60; Fig. 4b) account for much of it (5300 Å$^2$). The 160° bending induced by IHF binding is caused by the intercalation of two β-hairpins (β3 and β4) within the DNA's minor groove (Fig. 4b)[19]. In the hub, IHFα reaches out to TraY1 (158 Å$^2$) while both its chains interact with the TraI VH$_{2B/2B-like}$ sub-domain (~400 Å$^2$). Furthermore, TraY1 provides a binding interface to TraI VH$_{2B/2B-like}$ (410 Å$^2$) (Fig. 4b). These interaction areas are not individually extensive but their cumulative effect on TraI VH serves to calibrate TraI VH$_{2A}$ domain against its dsDNA binding site on the TraY arm of the dsDNA hairpin between bp +61 and +74 (Fig. 4c, Supplementary Fig. 7b).

Hub 2, composed of the train of TraYs, is characterised by a large network of protein-protein interactions between TraY subunits and protein-DNA interactions that anchor the train to 3 binding sites on the dsDNA (Fig. 5a, Supplementary Fig. 8). TraY is an RHH (ribbon-helix-helix) protein, which are usually dimers. In contrast, F TraY is a monomer containing all the secondary structures and tertiary fold of the RHH dimer in one single polypeptide. Overall, the three TraY molecules bury 2833 Å$^2$ of surface area upon interaction with their binding sites (see details in Supplementary Fig. 8). Interaction between TraY molecules is also extensive. Remarkably, TraY appears to be able to use very different surfaces to mediate protein-protein contacts (details in Supplementary Fig. 8).

In Hub 3, the TraI TE domain is observed bound to its previously characterised ssDNA-binding site between -2 and +9[33]. However, in marked contrast with previous observations (Supplementary Fig. 9a, b), it is also seen to make extensive contacts with the dsDNA between +10 and +15 (Fig. 5b). The latter interface accounts for nearly half of the total TE-DNA interaction area in the relaxosome (1295 Å$^2$ TE-dsDNA versus 1365 Å$^2$ for TE-ssDNA interactions). Interactions details are described in Fig. 5b and Supplementary Fig. 9a, b. Interestingly, residues in the β9-αF loop (Fig. 5b) displaces the R-strand base (a C) of the CG + 9 base pair out of its pairing arrangement, resulting in only G$_{+9}$ of the T-strand being visible in the electron density. Thus, the TraI TE domain is able to melt/unwind the +9 base pair, an observation that prompted us to ask whether this ability to melt/unwind dsDNA extends beyond +9 (see below). Finally, the TE domain provides a small but stabilising interface that anchors TraI VH$_{2B/2B-like}$ sub-domain within interactions with the dsDNA near the ss/ds DNA junction (Fig. 5c).

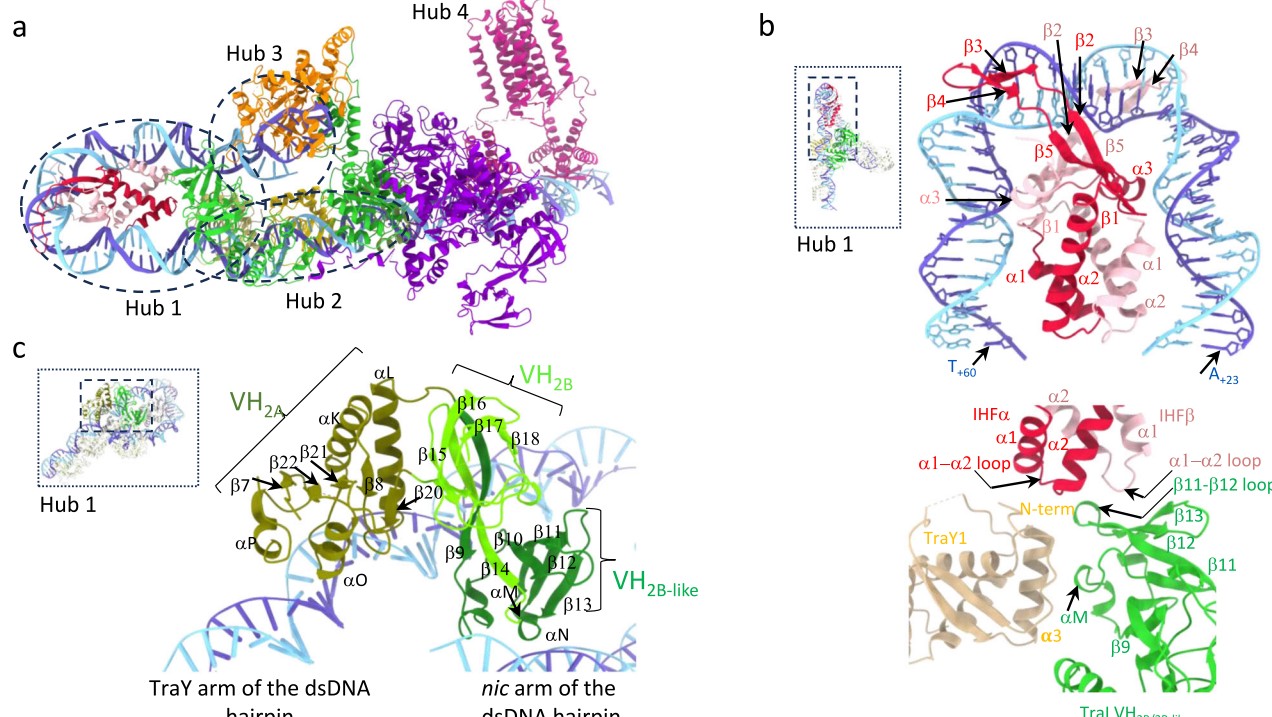

**Fig. 4 | Hub architecture of the ssDNA$_{-27\_+8}$ds$_{+9\_+143}$-R fully assembled relaxosome structure and overview of hub 1.** Proteins and DNAs are shown in ribbon representation. Secondary structures involved in DNA-binding are labelled. In panels **b**, **c** insets of the entire structure are provided for orientation. Within insets, dash-lined box shows the region(s) detailed in the corresponding panel(s). **a** Hub definition. The 4 hubs are shown within dotted lined ovals (except for hub 4) and are labelled hub1-4. **b** Hub1. Top: IHF-DNA interaction. Interactions are dominated by residues in the β2-5 region with bending triggered by insertion of the β2−3 hairpin into the major groove of the DNA (see Supplementary Fig. 7a, top panel, for details). Bottom: protein-protein interactions between TraY1, IHF and TraI VH$_{2A+2B/}$$_{2B\text{-like}}$ (see Supplementary Fig. 7a, bottom panel for details). TraY1 has a 12 residue

N-terminal tail, the very N-terminus of which is involved in interaction with residues in α1 and the α1-α2 loop of IHFα. The remaining N-terminal tail residues as well as many residues in its α3 helix interact with TraI VH. In TraI VH, residues in αM and the β9-αM and β11-β12 loops are implicated in TraY1 binding while other residues in β12, β13 and in the β11-β12 loop interact with residues in α2 and the α1-α2 loop of both IHFα and β. (see Supplementary Fig. 7a for details). **c** Interactions between TraI VH$_{2A+2B/2B\text{-like}}$ and DNA. Two regions of TraI VH are responsible for dsDNA binding at *oriT*: the tip of the 2B-like domain and a region of the 2A domain (details in Supplementary Fig. 7b). VH$_{2B/2B\text{-like}}$ also forms a three-way interaction in all complexes described here with IHF and TraY1 in hub 1, while also interacting with the TE domain of TraI in the ss$_{-27\_+8}$ds$_{+9\_+143}$-R complex in hub 3.

Next, to assess the magnitude of the conformational change needed in the T-strand to go from unbound to TE domain-bound, we in silico extended the ds$_{-27\_+143}$ DNA to include *nic* and superimposed it onto the ss/dsDNA of the ss$_{-27\_+8}$ds$_{+9\_+143}$-R structure (Supplementary Fig. 9e). We observe that not only has the *nic* site moved 29 Å away from its unbound location, but also that the T-strand on TE-binding has undergone a U-turn, imposed by its interaction with the TE domain and notably residues in the β10-β11 region (Fig. 5b). Thus, *oriT* is subjected to two topological U-turns, one in its dsDNA region upon IHF-binding, and another one in its ssDNA region upon TE-binding.

TraM is made of two domains (Supplementary Fig. 1f): an N-terminal domain (NTD) that interacts with DNA and a C-terminal domain (CTD) responsible for tetramerization and known to interact with the coupling protein TraD[35,42]. Thus, TraM can also be defined as a "hub" of protein-DNA and protein-protein interactions. This hub, Hub 4 (described in Supplementary Fig. 9c, d), is distinct from hubs 1-3 in that it not only mediates interactions between intra-relaxosome components but also with the T4SS[42].

An intriguing observation is that core relaxosome assembly is independent of the presence or absence of TraM as evidenced by the ss$_{-27\_+8}$ds$_{+9\_+143}$-RΔ$_{TraM}$ structure (Supplementary Fig. 4d, j) and the relative absence of interactions with other relaxosome proteins. Earlier deletion mutagenesis of the IHF, TraY and TraM binding sites (*IHFa*, *sbyA/C* and *sbmA-C* sites, respectively; Supplementary Fig. 1b) had shown that *sbm* sites were required for efficient DNA transfer, but not relaxosome formation, whereas *IHFa* and *sbyA* sites were required for both[34,43]. On the other hand, although not essential for the reaction,

TraM has been shown to stimulate TraI-mediated cleavage at *oriT* through interactions with the CTD of TraI[20,25]. This is consistent with our findings that TraM locates close to the CTD of TraI in the fully assembled ss$_{-27\_+8}$ds$_{+9\_+143}$-R relaxosome structure (Fig. 2e). In the complete absence of TraM binding sites, the frequency of DNA transfer is diminished 10,000-fold consistent with a key role for Hub 4[34].

A striking feature of the ss$_{-27\_+8}$ds$_{+9\_+143}$-R structure is the very large DNA footprint made by the various proteins involved (Fig. 5d). Protein-DNA interactions involve a staggering 9022 Å² of surface area, with IHF and the TraI TE domain accounting for more than half of this total. When the TraM/DNA interaction is added, over 10,000 Å² of buried surface area are involved in assembling the relaxosome components on *oriT*.

Many features of the F plasmid relaxosome assembly presented here apply to other plasmid systems, due to the TE domain of relaxases from multiple incompatibility groups such as the IncN, IncW, IncQ and other F-like IncF plasmids (includes R1, R100 and pED208) sharing a conserved fold (see above) (Supplementary Fig. 10)[33]. In R1 plasmid, the relaxase shares 92% identity with the F plasmid relaxase and has a similar domain organisation. TraY from R1 plasmid has a single RHH motif on a polypeptide and unlike F TraY forms a dimeric functional unit and may bind to the three subsites ATGT, ATTT (imperfect direct repeats) and ACAT (imperfect inverted repeat) on its *oriT* DNA. The consensus IHF binding sequence on R1 (5' TGATTTGCTATTTGAAT-CATTAACTTA 3') has a gap of 11 bp from the TraY1 binding site (ATGT), all of which points to a similar relaxosome organisation to that of F. Similarly, the F plasmid relaxosome proteins are able to bind and

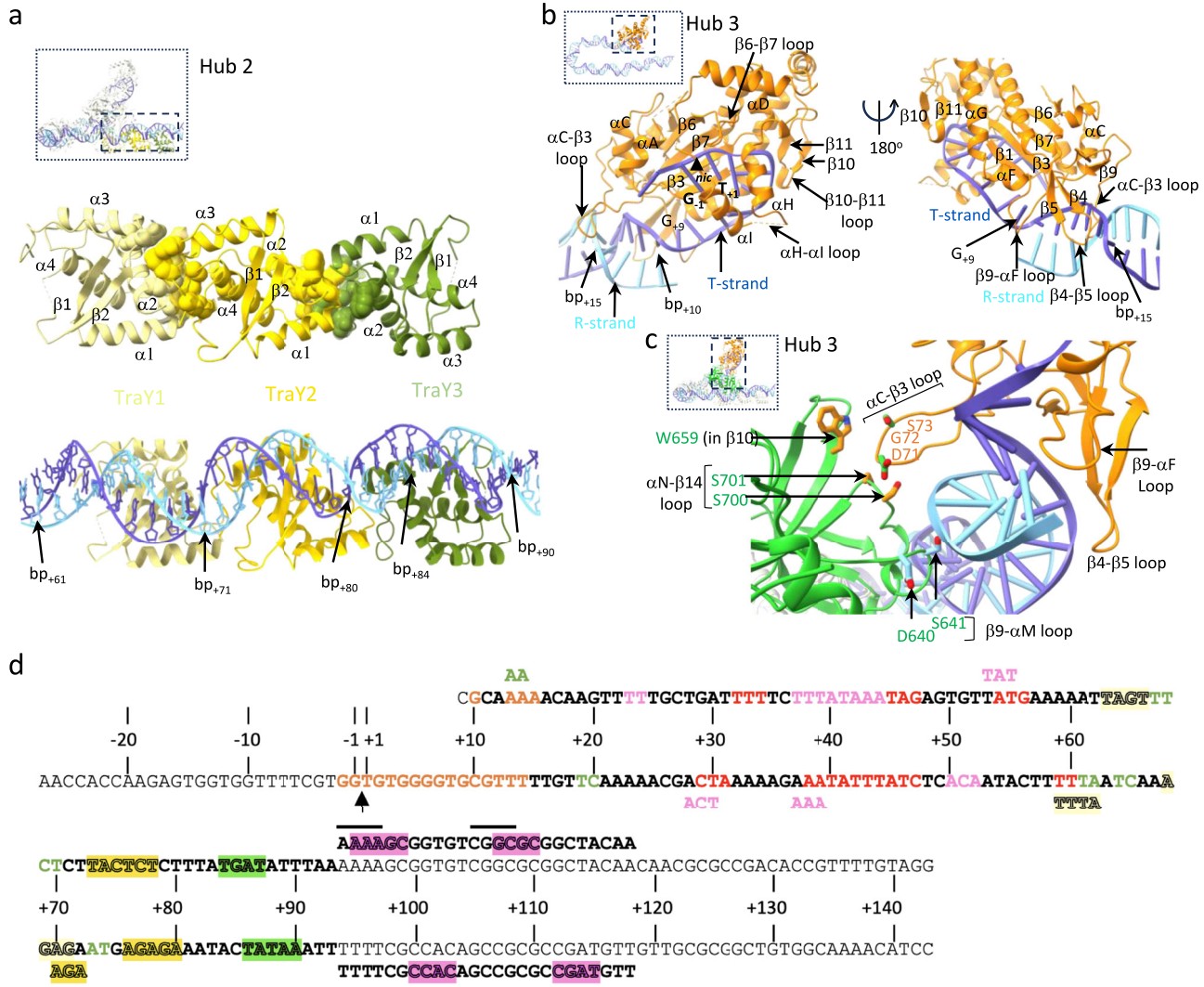

**Fig. 5 | Hub architecture of the ssDNA$_{-27\_+8}$ds$_{+9\_+143}$-R fully assembled relaxosome structure: overview of hubs 2 and 3, and footprint of relaxosome proteins on DNA. a** Hub2. Top: protein-protein interactions between TraY molecules. Residues at protein-protein interfaces are shown in spheres. Bottom: TraYs-DNA interactions. Boundary base pairs for each binding site are reported. TraY's binding to DNA is typical of RHH dimers with the β1 and β2 strands interacting with the major groove of the DNA binding site[79]. But binding of TraY1 and TraY2 (but not TraY3 which is inverted) also extends to the minor groove. TraY1 interaction with TraY2 involves two clusters, but only one is involved in interactions between TraY2 and TraY3 (detailed in Supplementary Fig. 8). **b** Hub3. TraI TE interactions with DNA. Left: view highlighting ssDNA-binding with primarily the β6-β7, αH-αI and β10-β11 loops but also dsDNA-binding, primarily with the αC-β3 loop. Right: view 180° away, highlighting dsDNA interaction with primarily with the αC-β3, β4-β5, and β9-αF loops. The *nic* site is shown as are the binding site boundaries on the DNA (ssDNA between -2 and +9 and dsDNA between +10 and +15). The unpaired +9 base is labelled. **c** Hub3 (continued): protein-protein interactions between TraI TE and

VH$_{2A+2B/2B-like}$ domains. TraI TE is shown in orange ribbon except the residues interacting with VH$_{2A+2B/2B-like}$ shown in green. VH$_{2A+2B/2B-like}$ is in ribbon and residues interacting with the TE and R-strand are shown in orange and light blue sticks. Residues and secondary structures involved in binding are labelled. One particular loop (αC-β3) stabilises 122 Å$^2$ of surface area via interactions with TraI VH$_{2B/2B-like}$ residues in β10 and the αN-β14 loop. **d** Footprint of relaxosome proteins on DNA. ssDNA and dsDNA bases involved in binding relaxosome components (including TraM, details in Supplementary Fig. 9c) are colour coded according to the protein they contact (red and pink letters for IHFαβ, green letters for TraI$_{VH2A+2B/2B-like}$, orange letters for TraI TE, and pale yellow, bright yellow, and olive green highlights for TraY1, TraY2, and TraY3, respectively, and magenta highlights for TraM). When two proteins have overlapping binding sites, the sequence is repeated above and coloured accordingly. The TraM footprint is shown above and under the R- and T-strand, respectively, in the *sbmC* region. Solid lines above the R-strand TraM site reports on previous findings based on hydroxyl radical protection assays.

conjugatively transfer chimeric *oriT* sequences combining the F plasmid *nic* site with the binding sites for TraY, IHF and TraM of plasmid R100[20].

The IncW prototype R388 plasmid encodes a relaxase, TrwC, lacking the VH domain of F relaxases. It has a TE domain required to produce the nick, an AH containing the ATPase and helicase activities, and a CTD (Supplementary Fig. 10a–d). The R388 relaxosomes are composed of the *oriT* DNA, the relaxase TrwC, the accessory protein TrwA, analogous in function to F TraM with an N-terminal RHH domain and C-terminal domain that interacts with the coupling protein

(termed TrwB in that system), and host protein IHF[44–46]. The R388 relaxosome therefore would be a smaller-scale assembly without the analogous TraY train. The AH of TrwC (structurally superposable to VH of F TraI) could sit across the IHF bend in a similar fashion, helping to position the TE domain close to the '*nic*' site (Supplementary Fig. 10e). DNA bending seems to be a recurrent theme in many relaxosome assemblies. An AlphaFold model of the RelSt3, a relaxase from the gram positive ICE*St3*/Tn*916*/ICE*Bs1* superfamily, along with putative relaxosome partners was proposed to bend DNA resulting in the catalytic domain of RelSt3 being positioned close to its *nic* site[47].

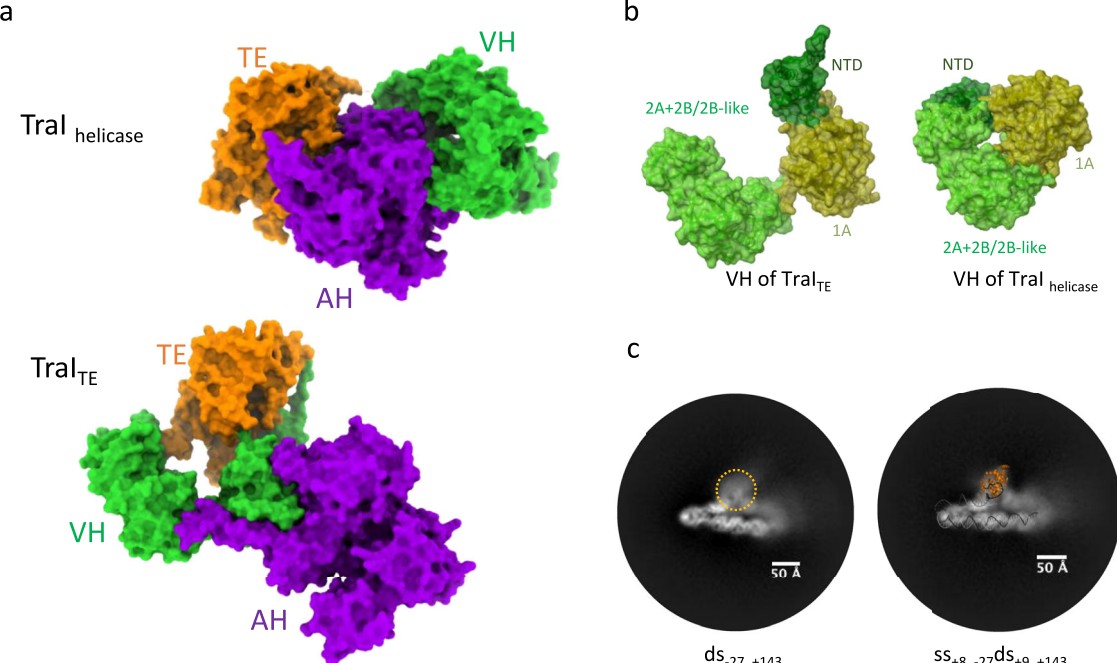

**Fig. 6 | TraI conformational changes, requirement for bubble formation 3′ of nic. a** TraI conformational changes. TraI TE, VH and AH domains are shown in surface representation, colour-coded orange, green and violet and labelled correspondingly. Top: TraI's helicase mode (TraI$_{helicase}$). Bottom: TraI's TE mode (TraI$_{TE}$). See main text for definition of TraI$_{TE}$ and TraI$_{helicase}$. **b** TraI VH domain in the relaxosome (TraI$_{TE}$ at left) versus in TraI$_{helicase}$ (right). VH$_{NTD}$, VH$_{1A}$ and VH$_{2A+2B/2B\text{-}like}$ are shown in dark green, olive and green surface, respectively. In TraI$_{TE}$, VH$_{NTD}$ and VH$_{2A+2B/2B\text{-}like}$ is found rotated by 47° and 95° relative to VH$_{1A}$, respectively. **c** The TE domain of ds$_{-27\_+143}$-R structure is flexibly positioned but locates in a region similar to that of the TE domain of ss$_{-27\_+8}$ds$_{+9\_+143}$-R. Two 2D-classes in the same orientation are shown, one for ds$_{-27\_+143}$-R (left) and one for ss$_{-27\_+8}$ds$_{+9\_+143}$-R. Left: a "fuzzy" density in shown in dash-lined orange circle. Right: the TE and DNA of ss$_{-27\_+8}$ds$_{+9\_+143}$-R is mapped on the 2D-class in such a way that its DNA is placed on top of the DNA's 2D-projection. This positions the TE in the density shown at left, demonstrating that this density is generated by the ds$_{-27\_+143}$-R TE domain. However, the density is "fuzzy", indicating the domain is flexibly located. Thus, TraI in ds$_{-27\_+143}$-R is likely similarly positioned as in TraI$_{TE}$.

## The TE and Helicase binding modes of TraI

In a previous study[17], we characterised two forms for TraI (Supplementary Fig. 1c, d). The first form was observed when TraI was bound to the ssDNA sequence 5′ of the *nic* site (*traI$_{TE}$* in Fig. 1b). It is sensitive to mild-proteolysis i.e. it adopts a flexible and "open" structure. In contrast, the second form is protease-resistant, i.e. rigid and "closed", and is observed only when TraI is bound to the ssDNA sequence 3′ of the *nic* site (*traI$_{helicase}$* in Fig. 1b). The structure of TraI bound to *traI$_{helicase}$* was solved[17]: in this structure, *traI$_{helicase}$* is bound through the helicase domains and this closed form of TraI was designated the TraI$_{Helicase}$ form of TraI.

With the ss$_{-27\_+8}$ds$_{+9\_+143}$-R relaxosome structure, we capture a state of TraI where the protein engages with its TE-binding site i.e. the sequence 5′ of the *nic* site. It is also very flexible since the AH and CTD domains of TraI remain largely semi-unstructured. It therefore captures the open form characterised previously[17]. We name this form of TraI "TraI$_{TE}$". In this conformational state, TraI has undergone a dramatic re-arrangement of its domain and subdomain structure compared to its helicase mode (Fig. 6a, b): i- while the AH domain was proximal to the TE domain in TraI$_{Helicase}$, in TraI$_{TE}$, it is the VH domain that is now proximal to that domain (Fig. 6a); ii- In TraI$_{TE}$, the VH domain itself has undergone a large re-arrangement of its sub-domains, with the NTD and the 2 A + 2B/2B-like module pivoting 47° and 95° degrees each relative to the 1 A sub-domain (Fig. 6b). Near atomic resolution was not achieved for the AH and CTD domains, nonetheless we were able to locate these domains within the relaxosome. Apparently, this region makes no specific and stabilising contacts with proteins or DNA nearby. Finally, when a model of the ss$_{-27\_+8}$ds$_{+9\_+143}$-R is placed onto the 2D-classes of ds$_{-27\_+143}$-R, we observe that the TE domain maps to a region of the 2D-class that forms a fuzzy protrusion from the dsDNA (Fig. 6c), indicating that this region of ds$_{-27\_+143}$-R, which has poor density and could not be interpreted, corresponds to the TraI TE domain. Thus, when in a dsDNA-bound relaxosome, TraI is in its TraI$_{TE}$ mode.

## TE binding requires the formation of a bubble at the helicase binding site

As mentioned above, in the ss$_{-27\_+8}$ds$_{+9\_+143}$-R structure, the R-strand base of the first double-stranded pair of the ss/dsDNA duplex (pair +9 in Fig. 5b) is not observed in the density due to its displacement by the β9-αF loop. Luo *et al.* determined that in vivo, DNA in the vicinity of the "*nic*" site showed increased sensitivity to KMnO4, indicating distortion/unwinding in this region[48]. We therefore asked whether TraI TE can melt a more extended region of dsDNA.

Thus, we reconstituted relaxosome complexes with decreasing ssDNA length, progressively extending the dsDNA part by increments of 5 nucleotides towards and beyond the *nic* site (Fig. 1c and Supplementary Fig. 2). The DNA employed were: ss$_{-27\_+3}$ds$_{+4\_+143}$, ss$_{-27\_-3}$ds$_{-2\_+143}$, ss$_{-27\_-8}$ds$_{-7\_+143}$ and ss$_{-27\_-13}$ds$_{-12\_+143}$. Cryo-EM datasets were collected for each complex. 2D classes and 3D maps were computed (see Methods and Supplementary Table 2). The ss$_{-27\_+3}$ds$_{+4\_+143}$-R and ss$_{-27\_-3}$ds$_{-2\_+143}$-R were solved to high resolution and are virtually identical to ss$_{-27\_+8}$ds$_{+9\_+143}$-R with DNA observed bound to a well-ordered TraI TE domain (Supplementary Fig. 4e, f, Supplementary Table 1a, b). For ss$_{-27\_-8}$ds$_{-7\_+143}$-R and ss$_{-27\_-13}$ds$_{-12\_+143}$-R, only 2D classes and initial low resolution 3D maps were obtained because it became clear early on in the processing that their structure was similar to that of ds$_{-27\_+143}$-R, i.e. with a disordered TraI TE domain (Fig. 7a, b). Thus, in the ss$_{-27\_+3}$ds$_{+4\_+143}$-R and ss$_{-27\_-3}$ds$_{-2\_+143}$-R complexes, the TraI TE domain engages with its binding site while, in the ss$_{-27\_-8}$ds$_{-7\_+143}$-R and ss$_{-27\_-13}$ds$_{-12\_+143}$-R complexes, it does not. Together, these results

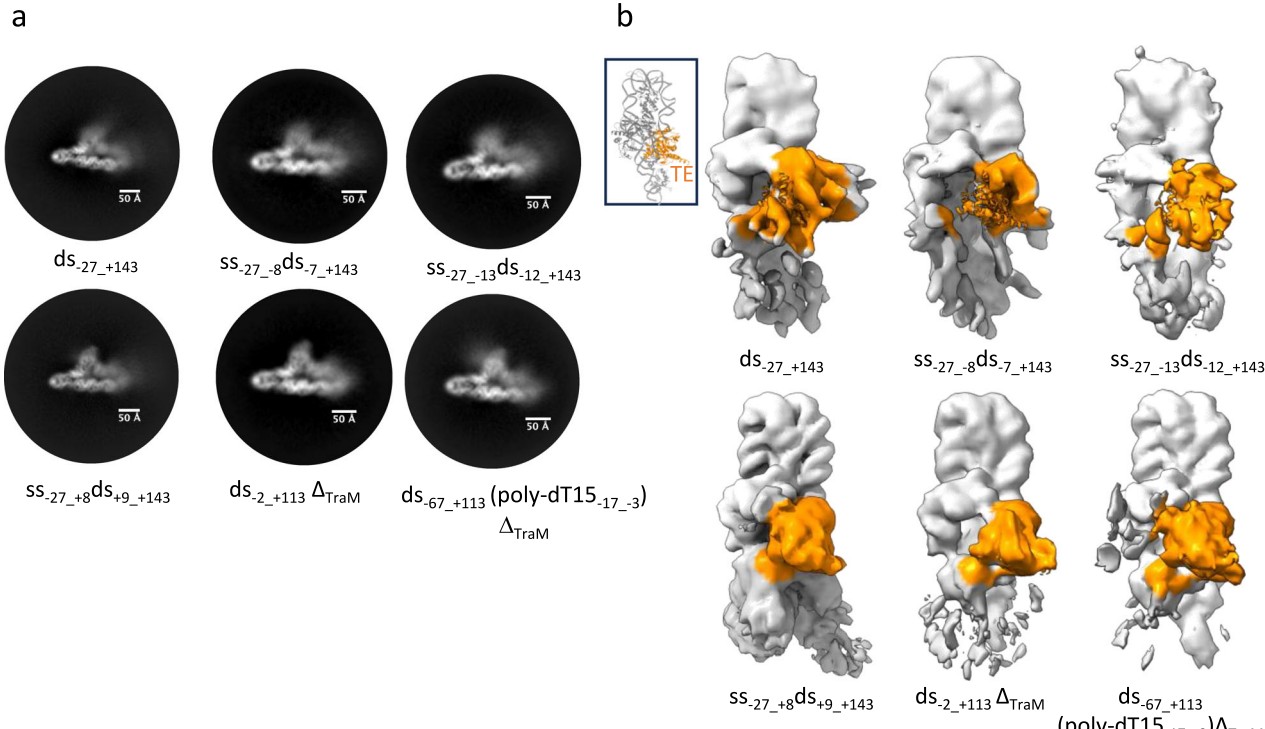

**Fig. 7 | Requirement for bubble formation 3' of *nic*. a** Requirement for a DNA bubble 3' of *nic*. Five 2D-classes are shown for five different relaxosome complexes reconstituted using five different types of DNAs. These DNAs are as described in Fig. 1c and Supplementary Fig. 2. Two types of classes are observed, regrouped into 2 rows. The first row reports on 2D-classes where the TE region is fuzzy, i.e. in those complexes, the TE domain is not bound to its binding site. The second row reports on 2D-classes where the TE region is well defined i.e. in those complexes, the TE domain is engaged with its binding site. See main text for description and inter-pretation of results. **b** Requirement for a DNA bubble 3' of *nic* (continued). Same as

panel a but reporting the corresponding 3D maps. The top row shows the 3D maps where the TE region is disordered: A map of similar resolution from initial pro-cessing of ds$_{-27\_+143}$-R dataset is shown for comparison. Bottom row shows the maps where TE is ordered, indicating its binding to the ssDNA. A map of similar resolution of ss$_{-27\_+8}$ds$_{+9\_+143}$-R is shown for comparison. Regions of the maps corresponding to the TE domain is coloured orange. See Methods section and Supplementary Table 2 for details. Inset shows the relaxosome structure in ribbon in the same orientation as the maps and with the TE domain coloured in orange.

indicated that TraI TE is unable to melt sequences beyond base pair -3. A corollary conclusion is that a single-strand of DNA is required before base pair -3 in order for TraI TE to melt and engage its binding site. To investigate this, we designed two experiments. Firstly, the relaxosome complex was reconstituted with a blunt end dsDNA of 115 nucleotides which include base pairs -2 to +9 (ds$_{-2\_+113}$) but does not extend beyond base pair -2 (Fig. 1c and Supplementary Fig. 2). A cryo-EM data set similar in size as for the 4 previous complexes described above was collected. 2D classes and 3D reconstruction maps show an ordered TE domain similar to those obtained for ss$_{-27\_+8}$ds$_{+9\_+143}$ (Fig.7a, b), indicating that a single-strand is not required to melt the TE-binding site, provided that the double strand does not extend beyond base pair -3. Secondly, we wondered whe-ther a bubble immediately 3' to the *nic* site would be sufficient for TraI TE to melt its DNA-binding site. To test this hypothesis, a dsDNA containing 15 bases (poly dT15) was inserted between position -17 and -3 as described in Fig. 1c or Supplementary Fig. 2. The DNA was extended on the 3' side of the T-strand to stabilise the bubble. TraM and its binding site were not included. This DNA is referred to as ds$_{-67\_+113}$(poly-dT15$_{-17\_-3}$). A limited cryo-EM dataset as above was collected, 2D classes and a 3D map were obtained and shown to resemble those of ss$_{-27\_+8}$ds$_{+9\_+143}$ indicating an ordered TE domain (Fig. 7a, b). Thus, a single strand bubble immediately 3' of *nic* is required for the binding reaction to occur.

## Validation of the relaxosome structure
To validate the structures presented here, we initiated an extensive mutational study, targeting residues in three hubs, Hub1-3 (Fig. 8a–d

and Supplementary Table 3). All mutations were made in *traI* and analyzed for conjugative plasmid transfer activity. Firstly, we targeted residues interacting with dsDNA (Fig. 8a). Both interactions of VH$_{2B/2B\text{-like}}$ residues with the *nic* arm of the dsDNA hairpin and of VH$_{2A}$ residues with the TraY arm were tested by generating negatively charged sub-stitutions. Double and triple mutants were made (Fig. 8a). We observed either no effect for residues of VH$_{2A}$ or very large increases in con-jugation rates for residues in VH$_{2B/2B\text{-like}}$ (nearly 300-fold for S641E-S696E for example), pointing to an interaction hub that plays impor-tant roles. Very similar results are obtained when mutating some residues at the interface between TraI VH$_{2B/2B\text{-like}}$ and TraY1 (Fig. 8b), between TraI VH$_{2B/2B\text{-like}}$ and IHF (Fig. 8c), and between TraI VH$_{2B/2B\text{-like}}$ and TraI TE (Fig. 8d). These results all emphasise the functional importance of the various hubs we have characterised in the structure. But perhaps more significantly, they point to the possibility that our structure represents a quiescent or pre-initiation state of the relaxo-some, the activation of which can be triggered readily by changes in the structure, possibly involving disruption of the key hub interactions analysed here. The activation step is predicted to occur in response to signals such as pilus termination or the presence of a recipient cell[49,50]. Our mutational study results indicate that changing any combination of residues disrupt a finely-tuned quiescent complex that can easily be triggered by extracellular signals.

## Mechanism of conjugation
A myriad of biochemical studies have demonstrated the relevance of in vitro reconstitution work from purified components in providing mechanistic insights on biological processes involving macromolecular

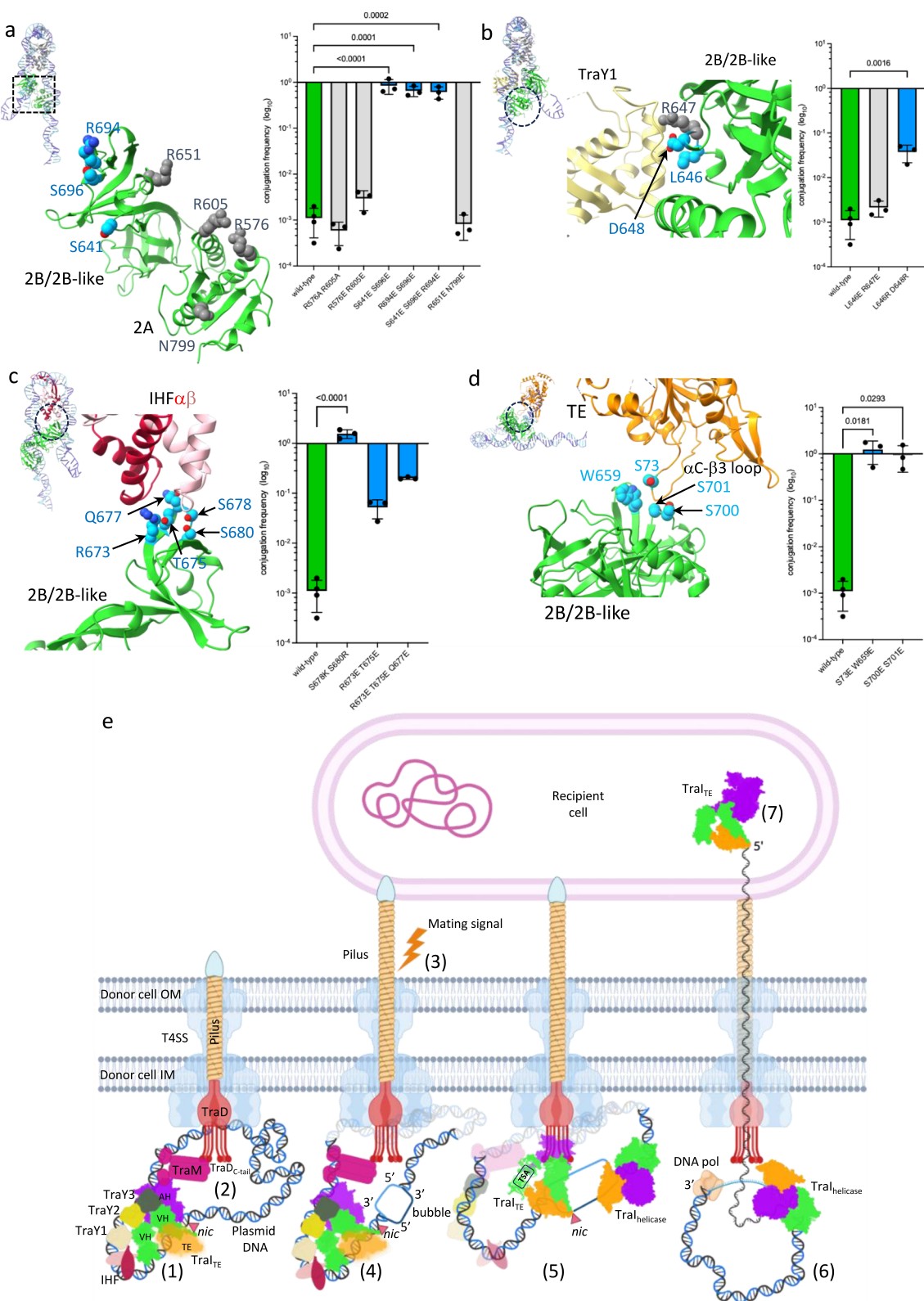

machines. Here, we present not only the structure of a relaxosome but also describe various DNA-bound states of this important complex. The structural characterisation of many more conformational states will be required to gain a full understanding of relaxosome function. Nevertheless, from the insights obtained here, a molecular-level mechanistic model for relaxosome recruitment and activation emerges (Fig. 8e). It starts with the formation of a quiescent relaxosome at *oriT* with TraI in its TraI$_{TE}$ mode (step 1) and its subsequent recruitment to the T4SS

(step 2) via the interactions of the coupling protein TraD with TraM and of TraI TSA and TSB modules with an unknown receptor[51,52]. Once pilus biogenesis is completed and a recipient cell has contacted a donor cell, a signal is sent (step 3) that activates the relaxosome by generating an ssDNA bubble' immediately 3' to the *nic* site on the T-strand (step 4). This allows i- the TE domain of TraI$_{TE}$ to access its binding site as in the structure of ss$_{-27\_+8}$ds$_{+9\_+143}$-R and ii- the loading of TraI$_{helicase}$[14] (step 5). The former functions in its so-called TraI$_{TE}$ mode when bound to *traI$_{TE}$*

**Fig. 8 | Mutational study of the relaxosome and Mechanistic model of conjugation. a–d** In each panel, the structure of the region of TraI where the mutations were introduced is shown in ribbon, with mutated residues in sphere representation colour-coded in blue or grey depending on whether they result in increased conjugation or not, respectively. Also reported is the log 10 bar graph of the conjugation rates observed for each of the mutant derivatives as described in the methods section. Data are presented as mean values +/− SD ($n = 4$ for wild-type, $n = 3$ for mutants). Colour-coding of bars is green, grey and blue for wild-type, mutants with no impact on conjugation, and mutants displaying increased conjugation, respectively. Ordinary one-way ANOVA with Holm-Šídák correction evaluated statistical significance between wild-type conjugation frequencies and complementation by mutant derivatives affecting each interface hub. P-values for each statistically significant difference ($P < 0.05$) is shown. Exact P-values for every comparison are indicated in Supplementary Table 3. Inset for each panel: zoom-out showing the entire relaxosome structure in the same orientation as main panel. **a,** TraI VH$_{2B/2B-like}$ interaction with DNA. **b,** TraI VH$_{2B/2B-like}$ interaction with TraY1. **c** TraI VH$_{2B/2B-like}$ interaction with IHF. **d** TraI VH$_{2B/2B-like}$ interaction with TraI TE. Source data are provided as a Source Data file. **e** The molecular details of conjugation in the donor cell. Shown are the outer and inner membrane (OM and IM, respectively), the T4SS (light blue), TraD (red), the pilus (yellow) and the relaxosome proteins TraM (magenta), TraY1−3 (pale yellow, yellow and olive green, respectively), IHF heterodimer (pink and red) and TraI$_{TE}$/TraI$_{helicase}$ (orange for TE, green for VH and violet for AH domain). The plasmid DNA is coloured light blue for the R-strand and dark blue for T-strand. '*nic*' site is indicated by an orange filled triangle. TE domain of TraI$_{TE}$ is shown in transparent orange to indicate conformational flexibility (not bound) or in solid orange (bound and ordered). (1) formation of a quiescent relaxosome, (2) relaxosome recruitment to the T4SS, (3) activating signal, (4) formation of ssDNA bubble on the T-strand, (5) loading of TraI$_{helicase}$, (6) DNA unwinding, (7) DNA is driven through the T4SS and the pilus, and injected into the recipient cell Created in BioRender[80].

site (Supplementary Fig. 1b), proceeds to cleave *nic* and attach covalently to the 5'phosphate resulting from the nicking reaction, while the latter when bound 3' of *nic* (*traI*$_{Helicase}$ site in Supplementary Fig. 1b) initiates DNA unwinding (step 6). In a final step, the DNA is driven through the T4SS, through the pilus, and finally injected into the recipient cell (step 7).

## Methods

### Cloning, protein expression and purification

All strains, plasmids, constructs, and primer sequences used in this study are listed in Supplementary Table 4a–d, respectively.

A tagless construct of IHF was generated by PCR amplification excluding the tag from a vector where *ihfA* and *ihfB* genes were cloned in tandem in the MCS of pRSF-1b plasmid (Novagen), to generate pRSF-IHF. The *traM* and *traY* genes synthesised and cloned into the NcoI and HindIII sites of pET28a vector were purchased from GenScript. All TraI expression constructs used in this study were engineered from pTrc99a-TraI[53] harbouring wild-type *traI* gene cloned into the MCS. Mutagenesis following the In-Fusion cloning protocol was used to generate the single Y16F mutant TraI variant (TraI$_{Y16F}$). Expression construct for TraI$_{Y16F}$ domain deletion variant TraI$_{\Delta AH+CTD}$ was generated by PCR amplification of pTrc99a-TraI, excluding the region of TraI between residue 835 to the C-terminus and annealing by In-Fusion seamless cloning technology (Takara Bio). All constructs were verified by DNA sequencing (Eurofins Scientific).

IHF was purified as a heterodimer with a protocol adapted from Nash et al.[54]. Briefly, *E. coli* BL21 (DE3) cells (ThermoFisher) transformed with pRSF-IHF were grown in 2 L of LB media supplemented with 50 μg/ml kanamycin at 37 °C. Upon reaching an OD$_{600}$ of 0.7, expression was induced with IPTG at a final concentration of 1 mM and incubated overnight at 18 °C. The cells were harvested by centrifugation, resuspended in 20 mM Tris-HCl pH 7.5, 100 mM NaCl (buffer A), one protease inhibitor cocktail tablet (Roche) and lysed by sonication (10–15 cycles of 5 s ON and 10 s OFF using a Sonics Vibra Cell VCX 130 sonicator set at 30% amplitude). The lysed cells were centrifuged, and the resultant supernatant was subjected to stepwise precipitation with ammonium sulphate to 50% and 70% saturation. The 70% precipitate was dialysed in 20 mM Tris-HCl pH 7.5, 50 mM NaCl (buffer B) and applied to a 5 ml HiTrap Heparin column (GE Healthcare) equilibrated with the same buffer. A gradient elution was performed using 50mM-1M NaCl in 20 mM Tris-HCl pH7.5, with fractions containing IHF heterodimer then concentrated and subjected to size exclusion chromatography using a Superdex 75 Increase 10/300 column (GE Healthcare) with buffer A as the mobile phase. The fractions containing the protein were concentrated, flash frozen in liquid nitrogen and stored in −80 °C.

TraM was purified as follows. C41 (DE3) cells (Lucigen) harbouring pET28a-TraM plasmid were grown in 1 L LB media with 50 μg/ml kanamycin at 37 °C. On reaching an OD$_{600}$ of 0.7, protein was induced by IPTG at a final concentration of 1 mM and incubated further at 18 °C, overnight. The pelleted cells were resuspended in buffer A with a protease inhibitor tablet (Roche). Lysis was carried out by sonication as described for IHF and the clarified lysate was subjected to precipitation with ammonium sulphate to 40% and 100% saturation. The 100% pellet was dialysed in buffer B and applied to a 5 ml HiTrap Heparin column (GE Healthcare). Elution was carried out using a gradient of 50mM-1M NaCl in 20 mM Tris-HCl pH7.5 with fractions containing TraM, detected by SDS-PAGE, pooled and diluted until the conductivity was equivalent to buffer B. This was applied to a 6 ml resource Q column (Cytiva) equilibrated with buffer B. A linear gradient of 50mM-1M NaCl resulted in the elution of proteins and the desired fractions were further diluted with buffer B. This was applied to a 5 ml HiTrap Blue column (Cytiva) and gradient elution was carried out with 50mM-2M NaCl. The fractions having the protein were then concentrated and subjected to size exclusion chromatography using a Superdex 200 Increase 10/300 column (GE Healthcare) equilibrated with buffer A. Fractions containing purified TraM tetramers were concentrated, flash frozen in liquid nitrogen and stored in −80 °C.

TraY was expressed in C41 (DE3) cells transformed with pET28a-TraY and lysed by sonication following the steps described for TraM. The clarified lysate was then loaded directly onto a 5 ml HiTrap Q column (Cytiva) connected in tandem to a 6 ml resource S column (Cytiva) such that the flow-through from the Q column passes through resource S. The Q column was then disconnected and gradient elution with 100mM-1M NaCl in 20 mM Tris-HCl pH7.5 was carried out on the resource S column. The fractions containing TraY were concentrated and further fractionated on a Superdex 75 Increase gel filtration column equilibrated with buffer A. The purified fractions were further concentrated and stored in −80 °C after flash-freezing in liquid nitrogen.

TraI, TraI$_{Y16F}$ and its truncation variant TraI$_{\Delta AH+CTD}$ were expressed in BL21 (DE3) cells (ThermoFisher) harbouring the corresponding pTrc99a clone, grown in 2 L of LB media supplemented with 100 μg/ml carbenicillin at 37 °C. Once the cells reached an OD$_{600}$ of 0.7, expression was induced with a final IPTG concentration of 1 mM and incubated overnight at 18 °C. The cells were harvested by centrifugation, resuspended in buffer A with one protease inhibitor tablet (Roche) and lysed by sonication as described above. The clarified cell lysate was subjected to ammonium sulphate precipitation (0.3 g/ml) with the resulting pellet dialysed in buffer B after resuspension in the same buffer. The solution after dialysis was applied onto a 5 ml HiTrap Heparin column (GE Healthcare) equilibrated with the same buffer. A linear gradient elution was performed using 50mM-1M NaCl in 20 mM Tris-HCl pH7.5 and the eluted fractions containing the protein diluted until conductivity was equal to buffer B. This was then applied to a 5 ml HiTrap Q column (GE Healthcare) equilibrated with buffer B. Gradient elution was carried out using 50mM-1M NaCl in 20 mM Tris-HCl pH7.5 and fractions having the protein were concentrated and subjected to

size exclusion chromatography on a Superose 6 Increase column equilibrated with buffer A. The purified protein fractions were concentrated and stored in -80 °C after flash-freezing in liquid nitrogen.

## DNA synthesis and purification

Synthesis of duplex $oriT_{316}$ DNA fragment was performed by large scale PCR amplification of the *oriT* region of F plasmid synthesised and cloned in pUC57 (pUC57-$oriT_{316}$), purchased from Genscript. A 5 ml PCR reaction containing 5 µg of template DNA, 0.5 µM final concentrations each of primers (ForiT_F, ForiT_R), 2X master mix of CloneAmp HiFi PCR Premix (Clonetech, USA) was carried out in a thermocycler with the following PCR conditions: 98°C for 2 min, and 30 cycles of 98°C for 10 s, 55°C for 5 s, 72°C for 5 s, and a final extension at 72°C for 10 min. The DNA was precipitated by addition of 1/10 volume of 3 M sodium acetate pH 5.2, 3 times volume of 100% ethanol and incubated o/n in -80 °C. Precipitated DNA was recovered by high-speed centrifugation, after which the supernatant was discarded. The pellet was dried at RT, washed with 100% isopropanol, further dried at RT and resuspended in 1 ml buffer A and a further step of purification was carried out on a Superose 6 Increase column equilibrated with buffer A. Fractions were assessed on a 4-20% TBE PAGE gel (ThermoFisher) after which the pure fractions were pooled giving a yield of 60-80 µg DNA, and stored in -20 °C.

For generation of the $ds_{-27\_+143}$ DNA by restriction digestion, a sequence containing five consecutive $ds_{-27\_+143}$ DNA fragments separated by six HindIII restriction sites, synthesised and cloned in pUC57-mini vector (pUC57mini- $ds_{-27\_+143}\_5X$), was purchased from Genscript. A large-scale restriction enzyme reaction mix containing 500 µg of plasmid, 10,000 units of HindIII-HF (NEB) and 1X CutSmart buffer was made up to a final volume of 12.5 ml with MilliQ water. The reaction was incubated at 37 °C for 1 h followed by heat inactivation at 80 °C for 20 mins and any aggregates were removed by centrifugation. The 12.5 ml mixture was concentrated to a volume of 100 µl in an Amicon Centricon Filter (Millipore) with a 3 kDa cut-off. The digested DNA was size-fractionated on a native-PAGE (5% TBE gel, 40% acrylamide/bis-acrylamide) using Mini Prep Cell (Bio-Rad). The fractions were inspected on a 4-20% TBE PAGE gel (ThermoFisher) and the ones containing $ds_{-27\_+143}$ DNA were desalted with buffer A and concentrated, in a 3 kDa cut-off Amicon Centricon Filter (Millipore), giving a yield of 50-60 µg DNA, and stored in -20 °C.

All the other DNA constructs used for relaxosome reconstitution were generated from PAGE purified Ultramer™ Oligos purchased from IDT. The corresponding R- and T- strand oligos (Supplementary Table 4e) were mixed at 1:1 molar ratio at a final concentration of 10 µM in a volume of 100 µl and annealed by incubating at 98 °C for 5 min, lowering the temperature at a rate of 1 °C/min until reaching 24 °C, in Annealing Buffer (50 mM TrisHCl pH7.2, 50 mM NaCl, 1 mM EDTA, 1 mM MgCl$_2$). The annealed DNA was purified by electrophoresis though a native acrylamide gel column (12% TBE gel, 40% acrylamide/bis-acrylamide) using Mini Prep Cell (Bio-Rad). The fractions were analysed for purity on a 4-20% TBE PAGE gel (ThermoFisher) and the pure fractions were desalted with buffer A and concentrated, in a 3 kDa cut-off Amicon Centricon Filter (Millipore), yielding 60-80 µg of DNA. Correct annealing of all annealed DNA heteroduplexes were verified by assessing the length of the two constituent ssDNA strands using a 10% TBE-Urea denaturing gel (ThermoFisher). The $ds_{-67\_+113}$(poly-dT15$_{-17\_-3}$) was additionally verified to confirm the unpaired region by S1 nuclease (Thermofisher) digestion to release its release right and left duplexes of 115 and 50 bp, respectively. The purified oligos were stored in -20 °C.

## Relaxosome DNaseI footprinting and sequencing of resulting fragments

The relaxosome complex was formed by mixing 0.25 µM $oriT_{316}$ DNA, molar excess of proteins at 8 µM each of IHF and TraM, 10 µM TraY, 5 µM TraI, 100 µl of 10X DNase I buffer (Roche) made up to a volume of

1 ml with Buffer A and was allowed to incubate for 30 min at RT. To this, 200 units of DNase I recombinant RNase-free (Roche) was added and further incubated for another 45 min at RT. Following this, an equal volume of premixed phenol:chloroform:isoamyl alcohol 25:24:1 (v/v) (ThermoFisher) was added, sample was mixed vigorously and centrifuged at 17,000 x $g$ for 10 min. The upper aqueous phase containing DNA was transferred to a new tube, mixed with 1/10 volume of 3 M sodium acetate pH 5.2, 3 times volume of 100% ethanol and incubated o/n in -80 °C. Precipitated DNA was recovered by centrifugation at 17,000 x $g$ for 30 min at 4 °C, after which the supernatant was discarded and the pellet was washed with 1 ml Isopropanol. The sample was centrifuged at 17,000 x $g$ for 30 min at 4 °C, supernatant discarded and the pellet was dried at RT. The air-dried pellet was resuspended in 100 µl MilliQ and purified by 1.5% agarose gel electrophoresis followed by extraction of DNA fragments from the excised gel band with Nucleospin gel and PCR Clean-up kit (Takara).

To facilitate cloning of DNA fragments into pCR-BluntII-TOPO vector (ThermoFisher), the fragments were subjected to blunting and dephosphorylation. For the blunting reaction 2 × 20 µl reactions each containing 1 µg DNA, 100 µM of each dNTP (NEB), 1 Unit T4 DNA polymerase (NEB), 1 × 2.1 Buffer (NEB) were incubated at 12 °C for 15 mins followed by heat inactivation at 75 °C for 20 mins. The reaction was purified with Nucleospin gel and PCR Clean-up kit (Takara) following the manufacturer's instructions for PCR clean-up. This was followed by dephosphorylation of the blunt ends, where 8 ×20 µl reaction volumes each containing 1 pmol DNA ends, 1 X CutSmart buffer (NEB) and 1 unit of Shrimp Alkaline Phosphatase (rSAP) (NEB) was incubated at 37 °C for 30 mins followed by heat inactivation at 65 °C for 5 mins. The reactions were pooled together and purified with Nucleospin gel and PCR Clean-up kit (Takara) according to the manufacturer's instructions for PCR clean-up.

A footprint library was generated by cloning the modified fragments into a pCR-BluntII-TOPO vector following instructions in the Zero Blunt TOPO PCR cloning kit (Invitrogen). Briefly, 6 µl reactions where DNA was mixed to achieve 1:10 or 1:20 molar vector-to-insert ratio, with 1 µl salt solution provided with the kit, were incubated at RT for 5 min. 4 µl of the reaction was used to transform Top10 competent cells (Invitrogen), plasmids were isolated from resultant colonies and the insert sequence established by DNA sequencing (Eurofins Scientific) (Supplementary Table 4f).

## Relaxosome complex formation, cross-linking and purification for cryoEM

A nickase inactive version of TraI harbouring Y16F mutation (TraI$_{Y16F}$) was used for all cryoEM complex formations[55] in order to prevent the forward reaction that might destabilise the complex. For cryoEM studies, complex formation was carried out as follows. To 0.25 µM *oriT* DNA supplemented with 20 mM MgCl$_2$, 20 times molar excess of purified proteins were added in the order of IHF, TraM and TraY followed by 5 times molar excess of TraI$_{Y16F}$ or TraI$_{\Delta AH+CTD}$ in a volume of 1 ml in Buffer A. This mixture was allowed to incubate for 30 min at RT following which the unbound proteins were separated from the DNA-protein complex on a Superose 6 Increase 10/300 column (GE Healthcare), equilibrated with 20 mM Hepes pH 7.5, 100 mM NaCl (Buffer C). The peak fractions containing the relaxosome were pooled (1 ml total volume) to which Glutaraldehyde (Grade I, 25% in H$_2$O, Sigma) was added to a final concentration of 0.5% to increase complex stability under cryo conditions. The cross-linking was allowed to proceed for 5 mins at RT and was terminated by the addition of TrisHCl pH 7.5 to a final concentration of 100 mM. The sample was subjected to a further round of gel filtration on a Superose 6 Increase 10/300 column (GE Healthcare), equilibrated with Buffer C, the fractions were pooled and concentrated using 30 kDa MWCO Microcon centrifugal filters (Millipore). The sample was analysed on a 4-12% Bolt Bis-Tris Plus gel (Invitrogen) and the gel band corresponding to the cross-linked

relaxosome were excised and constituent proteins identified by peptide mass fingerprinting where appropriate. The *oriT* DNA and protein components indicated in Supplementary Table 4g were used to form the different relaxosome complexes described in this study, but the protocol is as outlined above for each of the complexes.

## Negative stain EM

To check the quality and concentration of samples, 4 µl of 0.05 mg/ml cross-linked relaxosome was applied on glow discharged formvar-coated copper grids (Agar Scientific), incubated at RT for 1 min. The excess sample was blotted, and the grid stained with 4 µl of 2% uranyl acetate for 30 s. Excess stain was removed by blotting and the grids were allowed to dry at RT. The grids were imaged on a Tecnai T10 transmission electron microscope (Thermo Fisher Scientific) operated at 100 keV.

## CryoEM Grid preparation and data collection

For cryoEM analysis, 3 µl of the sample was applied to UltrAuFoil grids (Quantifoil, Germany; 2/2 200 mesh) that had been previously negatively glow discharged using PELCO Easiglow (Ted Pella, USA) and coated with a layer of graphene oxide[56]. The grids were incubated after sample application for 30 s in the chamber of a Vitrobot Mark IV (Thermo Fisher Scientific, USA) at 4 °C and 95% humidity and excess sample was blotted and vitrified in liquid ethane. The data were collected either at the ISMB Birkbeck EM facility or at the eBIC National facility using a Titan Krios microscope (ThermoFisher, USA) operated at 300 keV. The Krios was equipped with a post-GIF direct electron detector (Gatan,USA) and a BioQuantum energy filter (Gatan, USA) with a slit width of 20 eV. Images were collected automatically using the EPU software (ThermoFisher, USA) in super resolution mode with the total dose fractionated over 50 frames. The data collection parameters used for the different datasets are listed in Supplementary Tables 1a and 2.

## Image processing

The movies were first corrected for drift and dose-weighted using MotionCor2[57], followed by contrast transfer function (CTF) estimation using CTFFIND v4.1[58] within RELION[59]. Micrographs with a CTF-fit score better than 8 Å were selected for further processing. Workflows for image processing are reported in Supplementary Figs. 3 and 6.

**Image processing of ds$_{-27_{+143}}$-R.** A total of 55,914 movies were collected for ds$_{-27_{+143}}$-R from three different sessions giving three datasets which were processed separately and combined at a later stage. Initially, a subset of 4000 images were selected from dataset 1, to generate an initial model to test data quality and create templates for particle picking. From the 4000 images, 953,598 particles picked by reference-free auto-picking (Laplacian-of-Gaussian) in RELION were extracted with a box size of 400 pixels (1.06 Å/pixel sampling) and subjected to several rounds of 2D classification in cryoSPARC[60]. The resulting 374,358 particles were used for ab initio 3D classification followed by 3D refinement (cryoSPARC) of the best class (based on discernible features) to yield an initial 3D map (cryoSPARC). The map low pass filtered to 20 Å was used to obtain 30 randomized 2D projections, which were used as templates for reference-based Gautomatch v0.56[61] particle picking for datasets 1-3.

A total of approximately 3.5, 6 and 10 million particles were picked with permissive picking parameters for datasets 1, 2 and 3, respectively. Particles in datasets 2 and 3 were further split into smaller subsets for downstream classification steps to speed up processing. Several rounds of 2D classification in cryoSPARC resulted in a few high resolution 2D classes in a few preferred orientations but showing clear secondary structure features, and a majority of low-resolution classes comprising broken and junk particles. Selecting only the high-

resolution 2D classes resulted in maps with resolution anisotropy. Therefore, to avoid throwing away rare views the selection was kept liberal at the 2D classification stage, retaining 2D classes with any semblance of particle-like features and discarding only the 'obvious' junk classes. Instead, to tackle heterogeneity and clean up junk, we subjected the particles from 2D classification to rigorous rounds of ab initio 3D classification.

Best particles were selected from successive runs of ab initio classification where in each round the best 3D class was selected for the next round until an ab initio model with high-resolution features was obtained. The outcomes of datasets 1 and 2 were combined (556,418 particles) followed by 3D refinement in cryoSPARC resulting in a map of 5.97 Å. The map was used as an initial model for a round of alignment-free 3D classification in RELION with 6 classes and a regularisation parameter Tau of 16, to resolve further heterogeneity. Particles from the best two classes were combined with particles established from a similar workflow from dataset 3 and was refined to 4.31 Å resolution in cryoSPARC.

This consensus map of 337,238 particles showed disordered regions which disappeared when applying higher contour levels, indicating conformational heterogeneity. The density in the ordered region showing secondary structure details was subjected to local refinement in cryoSPARC with a mask excluding the disordered regions, yielding a map of 3.78 Å resolution. This map was used to build and refine models for *oriT* DNA, IHF, three molecules of TraY and TraI VH$_{2A+2B/2Blike}$. We could identify a poorly resolved region suggestive of 1 A of VH domain, but this region lacked secondary structure features and therefore no model was derived from it. The density in the disordered regions did not improve with further processing such as focused classification/refinement, particle subtraction etc.

**Image processing of ss$_{-27_{+8}}$ds$_{+9_{+143}}$-R, ss$_{-27_{+8}}$ds$_{+9_{+143}}$-RΔ$_{TraM}$, ss$_{-27_{+8}}$ds$_{+9_{+143}}$-RΔ$_{AH+CTD}$.** The overall strategy for the image processing of datasets with the ss/ds *oriT* heteroduplexes follows the one described above (Supplementary Fig. 6a–c). We were able to achieve high resolution maps with fewer images as compared to ds$_{-27_{+143}}$-R, since the samples seemed to have a higher degree of stability and homogeneity, likely due the binding of TraI TE which resulted in the arms of the DNA hairpin being less flexible. In the datasets of ss$_{-27_{+8}}$ds$_{+9_{+143}}$-R and ss$_{-27_{+8}}$ds$_{+9_{+143}}$-RΔ$_{TraM}$ which were collected from regions of thicker ice, more particles in diverse orientations were observed as evidenced by 2D classes and the angular distribution calculated from cryoSPARC heat maps (Supplementary Fig. 4b, d), yielding better 3D reconstructions in both cases.

For ss$_{-27_{+8}}$ds$_{+9_{+143}}$R, a final set of 155,077 best particles were selected, resulting in a global refined map of 3.77 Å resolution. The ordered regions showing secondary structure details in the global map were further subjected to local refinement yielding a map of 3.45 Å resolution. This locally-refined map was then used to build and refine models of the TraI TE domain bound to ssDNA, in addition to *oriT* DNA, IHF, three molecules of TraY and TraI VH$_{2A+2B/2Blike}$. Upon inspecting the global map, secondary structural features were clearly recognisable for the NTD and 1 A subdomains of VH and was used to dock AlphaFold 2 models for the same (see below).

In the global map of ss$_{-27_{+8}}$ds$_{+9_{+143}}$R we could also identify two poorly resolved regions, one next to the TraI VH domain and another at the distal end of the TraY arm of the *oriT* DNA. By comparing the global map from ss$_{-27_{+8}}$ds$_{+9_{+143}}$-RΔ$_{AH+CTD}$ (3.93 Å, 165,577 particles; Supplementary Fig. 4i) to the ss$_{-27_{+8}}$ds$_{+9_{+143}}$R map, we could assign this disordered region to the active helicase (AH) and C-terminal domains (CTD) of TraI. Similarly, the global ss$_{-27_{+8}}$ds$_{+9_{+143}}$-RΔ$_{TraM}$ map (3.11 Å, 330,708 particles; Supplementary Fig. 4j) was used to assign the density in the distal end of the *oriT* DNA to TraM and its DNA binding region.

**Image processing of ss$_{-27_{+3}}$ds$_{+4_{+143}}$-R and ss$_{-27_{-3}}$ds$_{-2_{+143}}$-R.** To further explore if the relaxosome could lead to localised melting of *oriT* near the *nic* site to facilitate TE docking, we extended the 5′ end of the R-strand by five and ten nucleotides to yield ss$_{-27_{+3}}$ds$_{+4_{+143}}$-R and ss$_{-27_{-3}}$ds$_{-2_{+143}}$-R complexes, respectively. The data collection and processing were essentially similar to that of the previous ssds constructs with the final locally refined map resolved to 3.68 Å (159,080 particles; Supplementary Fig. 4e) for ss$_{-27_{+3}}$ds$_{+4_{+143}}$-R and 3.42 Å (177,743 particles; Supplementary Fig. 4f) for ss$_{-27_{-3}}$ds$_{-2_{+143}}$-R.

**Image processing of ss$_{-27_{-8}}$ds$_{-7_{+143}}$-R, ss$_{-27_{-13}}$ds$_{-12_{+143}}$-R, ds$_{-2_{+113}}$-R, and dsR$_{-67_{+113}}$(poly-dT15$_{-17_{-3}}$).** It was evident from the processing of earlier datasets that the docking of TE domain indicating localised melting of *oriT* could be visualised from high resolution 2D classes and was observable even in initial 3D maps obtained from ab initio models in cryoSPARC. Therefore, for the next constructs tested for *oriT* melting, the data processing was carried out only till we obtained initial 3D maps. Briefly, pre-processing of the datasets were carried out as above, followed by 2D classification and a round of ab initio classification with refinement of the best class yielding low resolution 3D maps. For comparison, initial 3D maps obtained during early processing steps of ds$_{-27_{+143}}$-R and ss$_{-27_{+8}}$ds$_{+9_{+143}}$-R at a similar resolution are shown (Fig. 7a, b). The data collection statistics, resolution and initial and final particles obtained during processing are given in Supplementary Table 2.

All maps used for model building were sharpened using DeepEMhancer[62] and local resolution estimation was performed using cryoSPARC.

## Model building and refinement

Atomic model building into the 3.78 Å ds$_{-27_{+143}}$-R locally refined map was initiated by fitting the crystal structure of the IHF-DNA module (PDB: 1IHF) using UCSF Chimera[63]. Regular B-form DNA generated by Web 3DNA[64] was used as an initial model for the DNA arms and modified to fit the trajectory and helical pitch of the visible DNA density, using all-atom refine in COOT v0.9.8[65] with ProSMART restraints[66]. Three copies of a model of TraY generated using AlphaFold2 (AF)[67] (Supplementary Fig. 5a–d) were fitted into the three TraY densities visible in the cryoEM map using UCSF Chimera. AlphaFold2 predicted model of F TraI (Supplementary Fig. 5a–d) was similar to the experimental structure of its homologue R1 TraI in its helicase mode. From the F TraI AF model, the VH$_{2A+2B/2Blike}$ sub-domain structure was used for fitting into the corresponding density in the map. It should be noted that although the VH$_{2A+2B/2Blike}$ and AH$_{2A+2B/2Blike}$ superpose well, there are loops and secondary structure elements essential for ATP and ssDNA binding present in the latter, but absent in the VH$_{2A+2B/2Blike}$ sub-domain (Supplementary Fig. 5e). The complete model was then subjected to a few rounds of iterative manual rebuilding in COOT and refinement in PHENIX[68].

The low resolution of the map compounded by motion of the DNA arms as predicted from the 3D variability analysis in cryoSPARC, led to some ambiguity in the DNA base density. Although individual bases could be resolved, confident assignment and differentiating between purines and pyrimidines in the *oriT* density remained challenging. To assign the DNA sequence, we decided to take advantage of the three known TraY binding sub-sites described by the Schildbach group[69]. Therefore, starting from the three known TraY sites we manually updated the DNA sequence in COOT following the *oriT* DNA register. This assignment was only approximate, and the sequence was later reassigned guided by the ss/ds heteroduplex relaxosome maps which were of much higher quality in the DNA region (see below).

Further fitting of individual residues into the density were carried out by a combination of manual building using COOT and MDFF using ISOLDE[70]. The model was inspected for correct rotamer/Ramachandran outliers and clashes in ISOLDE. Real-space refinement of the

atomic model of ds$_{-27_{+143}}$-R was carried out in PHENIX with Ramachandran, secondary structure and nucleic-acid restraints applied.

To build the ss$_{-27_{+8}}$ds$_{+9_{+143}}$-R model into the corresponding 3.45 Å locally-refined map, the de novo built ds$_{-27_{+14}}$-R model was initially fitted into the density using UCSF Chimera. The crystal structure of the F TraI trans-esterase (TE) domain bound to its cognate ssDNA (PDB ID: 2A0I) was used as a model for fitting in the ordered TE density of the map. The higher resolution of the map and the stability of DNA arms resulted in an improved DNA density where the purines and pyrimidines were now clearly resolved. Therefore, the DNA sequence was accurately reassigned in the model manually using COOT. Individual protein and DNA residues were rebuilt into densities by iterative rounds of a combination of manual building in COOT and MDDF in ISOLDE. The model was inspected to correct rotamer/Ramachandran outliers and clashes using ISOLDE followed by refinement in PHENIX with restraints applied.

The completed atomic model for ss$_{-27_{+8}}$ds$_{+9_{+143}}$R without the TE-ssDNA was then used to further improve the ds$_{-27_{+14}}$-R model. After fitting into the ds$_{-27_{+14}}$-R map, it was rebuilt using COOT and ISOLDE with iterative rounds of real space refinement with restraints applied in PHENIX, to yield the final model for ds$_{-27_{+143}}$-R.

The ss$_{-27_{+8}}$ds$_{+9_{+143}}$-R model was also used as an initial model for the ss$_{-27_{+8}}$ds$_{+9_{+143}}$RΔ$_{AH+CTD}$, ss$_{-27_{+8}}$ds$_{+9_{+143}}$RΔ$_{TraM}$, ss$_{-27_{+3}}$ds$_{+4_{+143}}$R and ss$_{-27_{-3}}$ds$_{-2_{+143}}$R, rebuilt using COOT and ISOLDE with iterative rounds of real space refinement with restraints applied in PHENIX, to yield the final models for each of the complexes. These atomic models were nearly identical to ss$_{-27_{+8}}$ds$_{+9_{+143}}$-R with an average rms deviation of 0.738 Å for the Cα superposition.

For all the models, regions of poor Cα backbone density were subsequently deleted, and the side chains removed for areas with poor densities for side chains. The models were validated using Molprobity[71]. The statistics for model building, refinement and validation are reported in Supplementary Table 1b.

For fitting AF models of VH$_{NTD}$, VH$_{1A}$, AH, CTD and TraM (Supplementary Fig. 5a–d) to generate the ss$_{-27_{+8}}$ds$_{+9_{+143}}$-R global model, the unsharpened global ss$_{-27_{+8}}$ds$_{+9_{+143}}$-R 3.77 Å map was used. Initially, the refined ss$_{-27_{+8}}$ds$_{+9_{+143}}$-R structure was fitted in the global map. To generate a model for F TraM bound to DNA, an AF model of F TraM (Supplementary Fig. 5a–d) was first superposed onto the pED208 TraM:*sbmA* structure (PDB ID: 3ON0) with the CTDs as a guide. The NTDs of F TraM were adjusted to position on the NTDs of the DNA bound conformation of pED208 TraM to generate a model of F TraM bound to pED208 *sbmA*. This F TraM:pED208 *sbmA* model was docked as a rigid body into the map using UCSF chimera Fit in Map tool (cross-correlation 0.80 of map to model). The ends of the ss$_{-27_{+8}}$ds$_{+9_{+143}}$-R DNA proximal to TraM was extended by two base pairs, joined to pED208 *sbmA* DNA and the sequence was updated in the F TraM:pED208 *sbmA* model to match the *oriT* sequence by mutating DNA residues in COOT. From the AF generated model of F TraI, VH$_{NTD}$ (residues 307-385) and VH$_{1A}$ (residues 383-564) subdomains were separated and individually fitted in their densities in the map (correlation coefficients of 0.87 for VH$_{NTD}$ and 0.89 for VH$_{1A}$). Next, Isolde was used to flexibly refine these models with adaptive distance restraints implemented. The models were then merged with the VH$_{2A+2B/B-like}$ sub-domain to yield the complete VH domain.

For the AH domain, sub-domains of the F TraI AF model (Supplementary Fig. 5a–d) were fitted individually. The fitting was initiated with AH$_{1A}$ for which the density exhibited partial secondary structure features and the AH$_{1A}$ model (residues 934-1126) could be fitted with a correlation of 0.87. AH$_{NTD}$ (residues 835-933) was fitted in its density (correlation coefficient of 0.91) and both the models were flexibly refined in Isolde with adaptive distance restraints. Next the CTD (residues 1460-1630) model was rigid body fitted in the map with a correlation of 0.84, and flexible fitting was carried out in Isolde with restraints applied. The AH$_{2A+2B/B-like}$ model (residues 1127-1459) was

manually fitted in the density as this region was fragmented and poorly defined. The fitting was done such that the N-terminus and C-terminus of the $AH_{2A+2B/B-like}$ model was proximal to the C terminus of $AH_{1A}$ model and the N terminus of CTD, respectively. $AH_{NTD}$, $AH_{1A}$ and $AH_{2A+2B/B-like}$ models were joined to give the full AH domain. This gave us the composite global model of a fully assembled relaxosome. The models thus assembled gave rise to minimal clashes in the structure and the fitting of sub-domains as described above resulted in the corresponding N and C termini being in proximity such that they could all be merged confidently. Note that the side chains for $VH_{NTD}$, $VH_{1A}$, AH and CTD were removed in the final model.

### Interactions, analysis, visualisation and representation
Interactions were calculated using the PDBePisa server[72]. Domain movements were analysed using the DynDom program[73]. CryoEM maps and models were visualised and figures prepared using UCSF Chimera[63], UCSF ChimeraX[74] and PyMOL[75].

### Construction of p99I+ mutant derivatives
Double- or triple- site mutations in F traI were generated by cutting p99I+ with Eco0901 to remove a 1.4 kb fragment of TraI or in case of variant S73E W659 with KpnI/Eco0901 to remove 3 kb. Forward primer MutI fw and the reverse primer MutI rev were used to create all mutant alleles except mutant S73E W659E which required forward primer MutI17fw (Supplementary Table 4c). Mutant-specific primer pairs (Supplementary Table 4d) containing the mutations in their overlap region were used together with these forward/reverse pairs to generate fragments suitable for error free joining via seamless assembly of multiple DNA Fragments [NEBuilder HiFi DNA Assembly Kit (NEB)]. *E. coli* DH5α cells were transformed with the ligation products. Oligonucleotide synthesis and Sanger sequencing of the mutated variants was performed by Microsynth Austria.

### Bacterial conjugation
Conjugative plasmid transfer was performed as described in ref. 76. The conjugation deficient phenotype of *E. coli* MS411 [pOX38ΔtraI] donor cells was complemented in trans with wild-type [p99I+] or mutant traI alleles. Overnight cultures of donor strains were diluted to an optical density at 600 nm ($OD_{600}$) of 0.005 in 900 µl antibiotic-free LB medium and incubated for 1 h at 37 °C before 100 µl of recipient culture ($OD_{600}$ of 2) was added. Conjugation was allowed to proceed for 1 h at 37 °C and then stopped by vortexing for 1 min. Transconjugants were selected on plates containing 25 µg/ml Sm and 40 µg/ml Km. Conjugation frequencies were expressed as transconjugants per donor cell (Supplementary Table 3).

### Statistical analysis
Statistical analysis was performed using GraphPad prism version 10. Comparisons of conjugation frequencies of wild-type TraI and mutated variants for each interface hub was performed by ordinary one-way ANOVA with Holm-Šídák's multiple comparisons test; $P < 0.05$ was considered significant (Supplementary Table 3).

### Reporting summary
Further information on research design is available in the Nature Portfolio Reporting Summary linked to this article.

## Data availability
EM maps and atomic models were deposited to the Electron Microscopy Data Bank (EMDB) and Protein Data Bank (PDB) databases. Accession codes can be found in Supplementary Tables 1 a, b and 2. PDB codes for the various structures reported in this manuscript are 9F0X, 9F0Y, 9F0Z, 9F10, 9F11, 9F12 and the EMDB accession codes are EMD-50117, EMD-50118, EMD-50119, EMD-50120, EMD-50121, EMD-50122, EMD-50131, EMD-50132, EMD-50133, EMD-50098, EMD-50099, EMD-50102, EMD-50103, EMD-50104, EMD-50105 and EMD-53548. A model of the fully-assembled ss-$_{-27\_+8}$ds$_{+9\_+143}$-R relaxosome and an associated ChimeraX session can be found with the Supplementary Data 1. All constructs (wild type and mutants) used in this study can be obtained on request to GW. Source data are provided with this paper.

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

## Acknowledgements

We would like to thank Dr. David Houldershaw for IT support and Drs. Natasha Lukoyanova and Shu Chen for collection of data on the in-house Krios. This work was supported by Wellcome grants 098302 and 217089 to GW and the Austrian Science Fund (FWF) [10.55776/DOC50 *Molecular Metabolism*] to ELZ. Cryo-EM data were collected at the ISMB EM facility at Birkbeck College, University of London with financial support from Wellcome Trust (202679/Z/16/Z and 206166/Z/17/Z) and at the UK national electron Bio-Imaging Centre (eBIC).

## Author contributions

S.M.W. purified the DNA and protein components, established the conditions for relaxosome assembly and set up the DNaseI footprinting protocol. S.M.W. conducted NS and cryoEM sample preparations, data collection, processing, model building and refinement. SMW and GW carried out structural analyses. S.R., S.K., and E.L.Z. performed the mutational analysis of *traI*. AI was involved with relaxosome characterisation in the initial phase of the project. S.M.W., E.L.Z. and G.W. wrote the manuscript with inputs from SR. GW supervised the work.

## Competing interests

The authors declare no competing interests.
