## [Transparent Peer Review file · Nature Communications]

Cryo-EM Structure of the relaxosome, a complex essential for bacterial mating and the spread of antibiotic resistance genes

Corresponding Author: Professor Gabriel Waksman

Version 0:

Reviewer comments:

Reviewer #1

(Remarks to the Author)

In this article, Williams et al describe a series of structures that represent the relaxosome complex of the plasmid F bound to the oriT region in various states. These structures provide the most complete overview of the relaxosome complex up to date. They give important insights into the function and mechanism of the distinct components of the complex. The authors propose a series of mechanistic snapshots based on the structures, presented in Fig 4e.

The manuscript describes a large body of work that represents a big step forward in the structural description of the relaxosome of plasmid F and has the potential to inspire many follow up experiments to corroborate and expand the proposed mechanism. It is well written and the validation and conclusions are appropriate. I therefore think that the scientific contribution of the paper will be of great value in the field of conjugation of plasmid F.

As a critical note, the paper does not refer back to previous work and therefore lacks context. I think that this is important to do, because many conjugational systems exist, each with its particularities concerning the organization of the molecular functions over the different components. Without this context, this work therefore seems of limited interest to groups working on systems other than plasmid F. For example, many structures exist of the relaxase DNA binding domain, not just of Tral, but also for the related TrwC from R388 and from relaxases more distantly related. I suggest a comparison is made between these structures and the TE domain of Tral in the ss-27_+8ds+9_+143-R and ss-27_+8ds+9_+143-R structures.

As a second example of possible context, there is the work published in 2017 by the same group on the same helicase, where they suggest dimerization is necessary for a functional helicase and present the structure of a ssDNA bound to FL Tral. That work is cited in the manuscript, and the mechanism is discussed in the introduction, but the integration of the structures published here and of the previous work is not done (I am thinking of Fig S7 of the previous work for example). I think the paper would be enriched by this analysis.

The structural detail on the conformational change in the relaxase to go from the TE to the relaxase mode is in my opinion one of the nice features of this work. This change is induced by binding of the TE domain to ssDNA. Apparently, the TE mode is not compatible with the structure of the full relaxosome? I think the readers would be very interested in a description of the allosteric mechanism that causes the conformational transformation of the VH domain.

One of the revelations of the full relaxosome structure is that the relaxase spans two DNA arms on both sides of the IHF binding site, using the TE and the helicase domains respectively. However, the C-terminal domain of many other helicases are highly variable and many relaxase families lack helicase domains at their C-terminus. Therefore, this structure suggests that relaxosomes in other systems may have a completely different structure.

As I stated above, I think this work will spark many follow up studies as it raises some interesting points on the relaxosome assembly and function. For example, TraM is often included as a relaxosome protein, but the current work suggests that it is much less involved in the relaxosome complex as there are very few interactions with the Tral, TraY, IHF proteins and their DNA loci. Does this result fit previous consensus or is it a new insight? Related to this, how accurate is the TraM C terminal position, considering that the B factors are very high, consistent with the linker connecting N- and C-terminal is very flexible

and there are no interactions?

Some minor comments:

- Why is the Tral sequence cut up into different chains in the pdb structure? I suggest that the same chain identifier be used for all residues of the Tral
- The names used in the paper for the different structure should be included in fig 1c-e
- Lines 231 to 234, "The AH and CTD domain ... ie is semi flexible" should be revised.
- The use of TralTE as mode and Tral TE as domain can be confusing. Eg, in line 239, specify that this refers to the domain

Reviewer #2

(Remarks to the Author)

Remarks to the Author:

Conjugation is an important research topic due to its role in the spread of antibiotic resistance. The process involves a type 4 secretion system (T4SS) that spans the double membrane, a conjugative pilus, and a protein complex called the relaxosome, which binds to and processes the plasmid before it is transferred to a new host cell. While recent research has provided insights into the structure of the T4SS and pilus, the structure of the relaxosome has not been resolved. The present study by Williams et al fills this gap by reporting the cryo-EM structure of the fully assembled relaxosome encoded by the paradigm F plasmid. The authors report two distinct complex states, which they suggest represent different functional steps in plasmid DNA processing. These findings are intrinsically novel and contribute valuable new information to the field, summarized in a comprehensive model (Fig. 4) for conjugation steps. However, the manuscript would benefit from a more accessible presentation, a deeper discussion of mechanistic insights and their biological implications, as it currently lacks a clear narrative to help guide the reader and highlight the significance of these new structures.

While the the experimental approach is logic and the structural insights well-articulated, more discussion on how these findings differ or align with existing knowledge in relaxosome assembly and binding to the oriT would strengthen the manuscript's impact. The manuscript would benefit from a comprehensive discussion on whether the structural findings are likely unique to the F plasmid or could be (even partially) generalizable to other plasmids.

The study provides a detailed structural breakdown of the relaxosome's hubs, but the functional significance of these hubs could be clarified and expanded. The manuscript emphasizes the flexibility of certain relaxosome components, particularly Tral helicase and trans-esterase domains, but this point could be explored further. The manuscript shows that mutations in specific residues within the protein-DNA and protein-protein interaction hubs either disrupt or enhance conjugation efficiency. Providing a more detailed explanation of why certain residues are particularly critical (e.g., conserved catalytic sites in Tral) would solidify the structure-function relationship. The role of DNA bending and unwinding (e.g., U-turns induced by IHF and Tral) in relaxosome assembly is described. How critical are these bends to the precise positioning of the "nic" site and subsequent DNA unwinding?

The methodology is thorough but highly technical. A summarized explanation of cryo-EM's relevance and why it was chosen over other structural techniques would improve clarity. Figures are well-integrated and detailed. However, descriptions could be simplified in figure legends to make them more self-contained for readers less familiar with the methods. By addressing these areas, the manuscript would not only enhance its depth and breadth but also increase its appeal to a broader audience, from structural biologists to researchers in antimicrobial resistance

Reviewer #3

(Remarks to the Author)

This manuscript presents the first high-resolution structural analysis of the bacterial relaxosome, a nucleoprotein complex involved in bacterial conjugation. Using cryo-electron microscopy and modeling, the authors reveal detailed structures of the fully assembled F plasmid relaxosome in its pre-activation state, providing new structural insights into bacterial DNA transfer machinery. Given that plasmids are major carriers of antibiotic resistance genes, this work has direct implications for understanding resistance spread.

The relaxosome consists of three plasmid-encoded proteins (Tral, TraY, and TraM) and one host-encoded protein (IHF) assembled on a specific DNA sequence called oriT (origin of transfer). The authors determined the structure of the relaxosome in its pre-activation state. Using different DNA constructs, they investigated how the complex recognizes and processes its DNA substrate. The structure reveals an intricate organization where the DNA forms an asymmetric U-shaped hairpin, with proteins arranged in four distinct "hub" regions that combine protein-DNA and protein-protein interactions. The central protein Tral is observed in a 'TE mode' (trans-esterase mode) conformation, which differs significantly from its previously known 'helicase mode' structure.

The structural analysis reveals extensive interaction networks between the protein components and DNA. The authors mapped multiple protein-protein interfaces and protein-DNA contacts, providing atomic-level detail of how the complex is

assembled. A particularly notable feature is the DNA architecture, which includes two dramatic U-turns: one in the double-stranded DNA region induced by IHF binding, and another in the single-stranded region upon Tral TE binding.

Mutational analysis of Tral interfaces was performed, showing that disrupting certain protein-protein interactions affects conjugation rates. Based on their structural observations, the authors propose a mechanistic model for bacterial conjugation, though this model would require additional functional validation.

This work provides the first detailed structural framework for understanding relaxosome assembly. While the functional implications of these structures remain to be fully explored, they offer a foundation for future studies of bacterial conjugation mechanisms and potentially inform strategies to combat antibiotic resistance spread.

The study has several important limitations to consider:

The functional data in the paper is quite limited. Conjugation frequency assays for Tral mutants (Fig. 4a-d), measuring transfer rates when key interface residues are mutated. While this provides some functional validation of the structural observations, these are relatively basic assays focused only on Tral mutations. DNase I footprinting experiments (Extended Data Fig. 2) to map protein binding regions on oriT DNA. However, this is more of a structural validation rather than true functional analysis. The manuscript lacks several types of functional studies that would strengthen its conclusions, such as direct biochemical assays showing the proposed DNA bubble formation and its role in activation. While these additional experiments would have strengthened the mechanistic model, they are probably beyond the scope of this structural study.

While the structural work is impressive, the manuscript misses opportunities to integrate these new findings with the extensive existing literature. Previous biochemical and genetic studies had identified key residues in Tral, characterized IHF's role in DNA bending, and mapped essential regions of oriT - yet there is limited discussion of how this structure explains or challenges these findings. Similarly, while the authors cite structures of relaxases from Gram-positive bacteria and mention the R1 plasmid system, they don't provide detailed structural comparisons that could have provided broader mechanistic insights into bacterial conjugation systems.

The complex was reconstituted in vitro from purified components rather than isolated from cells and chemically cross-linked with glutaraldehyde, which could potentially trap non-physiological conformations. The authors used a catalytically inactive mutant (Tral Y16F) which might affect the native structure. The bias or limitations of using a mutant, in vitro reconstitution and glutaraldehyde crosslinking was not explicitly discussed in the paper.

Another limitation of this study is that it primarily captures a single "quiescent" pre-activation state of the relaxosome, leaving critical questions about the complex's dynamic behavior unanswered. The absence of structures representing the active state, particularly after cell-cell contact triggers conjugation, creates a significant gap in our mechanistic understanding. Furthermore, several functionally important domains, including the Active Helicase (AH) and C-Terminal Domain (CTD), remain partially disordered in the structures, limiting our insight into their roles during DNA processing and transfer. This could be discussed.

The study relies heavily on computational approaches and modeling to generate its structural conclusions. The authors used AlphaFold2-predicted models for fitting several components into their cryo-EM maps and employed extensive computational modeling to interpret poorly resolved regions. Notably absent are any quality metrics for these AlphaFold2 models, such as pLDDT scores, PAE plots, making it difficult to assess the reliability of the computational models used for structural interpretation.

Despite these limitations, which are often inherent in challenging structural biology projects, this work represents a significant advance in our understanding of bacterial conjugation machinery.

Version 1:

Reviewer comments:

Reviewer #1

(Remarks to the Author)

The authors made an effort to address on my comments, which I appreciate. I apologize if my previous comments were not clear. They were made because I find the conformational rearrangement of Tral in the TE and helicase mode remarkable and believe that this deserves more attention than that given in the current manuscript.

The VH domain relocates from the exterior of the protein in the Tral(helicase) mode, to the boundary between the TE and AH domains in the Tral(TE) mode. In addition, the VH domain itself undergoes a large rearrangement (Fig 5b). Page 13 line 284-285 state that this part of the protein does not form interactions with nearby molecules and page 13, lines 282-283 state that the AH and CTD domains are largely unstructured. It is therefore not clear what brings about this change and the manuscript could clarify better on what basis the VH and AH were built like this. In particular, I hope the authors can address the following points:

- Fig 5A and Fig S1 only shows a schematic representation of the Tral(TE) and the Tral(helicase) structures, but there are no density maps of the Tral(TE) mode, or superpositions of the domains of the two structures in the current manuscript. I suggest to include in a supplementary figure i) density maps of the regions mapping to the individual domains of Tral(TE), ii)

superpositions of the TE and AH models in TE and helicase modes, where possible, to map the changes in these domains that allow the relocation of the VH domain.

- Given the resemblance between the structures of the VH and AH (Langovan et al. 2017 page 714), I wonder if the density around the VH and AH is sufficiently clear to distinguish between the two, can the authors comment on that? Perhaps show clear evidence of density in regions that help distinguish between the VH and AH (see also my previous point)?

- I would suggest an analysis of the possible allosteric pathways for the transition between the Tral(TE) and Tral(helicase). What conformational changes in the protein allow for the relocation of the VH? How are the interactions surfaces between the TE domain and the AH domain with the VH domain in TE mode? Have similar conformational rearrangements been described for helicase domains relate to VH?

Concerning the added Suppl Fig 9, showing the superposition of R388 TrwC AH with the Tral VH of the Tral in TE mode: is the TrwC AH domain in open (aka TE mode) or closed form (aka helicase mode)?

I hope that the authors and editor agree that these suggestions would strenghten the paper.

Minor comments:

The 2017 paper describes an open and a close conformation of the relaxase. I suggest the authors explicitly link the terms "open" and "closed" state, used in the 2017 cell paper, to the different states reported in this manuscript.

page 16, line 368: change "the TE of TralTE" to "the TE domain of TralTE"

page 9, line 198: change "Tral TE is observed" to "Tral TE domain is observed"

page 10, line 212 "from unbound to TE-bound": clarify that the TE domain is meant here

page 10, line 216: change "imposed by its interaction with TE" to "imposed by its interaction with the TE domain"

page 11, linne 245: change "with IHF and Tral TE" to "with IHF and the Tral TE domain"

Reviewer #2

(Remarks to the Author)

The authors have adequately addressed all major concerns raised by this reviewer, particularly by emphasizing how their findings integrate into existing knowledge and by discussing the broader implications of their results for other plasmid systems. Most of the reviewers' comments centered on reorganization of the manuscript and deeper discussion of the data, rather than necessitating additional experiments or new analyses. The reviewer also recognizes that some suggestions—specifically the concerns raised in Comment 4—fell outside the scope of this study and therefore could not be fully addressed. Nevertheless, the authors' revisions successfully clarify and improve the quality of the manuscript, which now appropriately reflects the high quality of the experimental work

Reviewer #3

(Remarks to the Author)

The authors have addressed all my comments and suggestions

made.

8th March 2025

Dear Editor

Please find enclosed a revised version of our manuscript entitled “Cryo-EM Structure of the relaxosome, a complex essential for bacterial mating and the spread of antibiotic resistance genes”. The reviewers’ comments are overwhelmingly positive. Reviewer 1 states: “These structures provide the most complete overview of the relaxosome complex up to date. They give important insights into the function and mechanism of the distinct components of the complex”. Reviewer 2 states “These findings are intrinsically novel and contribute valuable new information to the field, summarized in a comprehensive model (Fig. 6) for conjugation steps”. Finally, reviewer 3 states that “This work provides the first detailed structural framework for understanding relaxosome assembly. While the functional implications of these structures remain to be fully explored, they offer a foundation for future studies of bacterial conjugation mechanisms and potentially inform strategies to combat antibiotic resistance spread”.

However, all reviewers have suggestions to improve the paper. We have found these suggestions extremely useful and have incorporated them as detailed below.

Reviewer #1 (Remarks to the Author):

Comment 1:

In this article, Williams et al describe a series of structures that represent the relaxosome complex of the plasmid F bound to the oriT region in various states. These structure provide the most complete overview of the relaxosome complex up to date. They give important insights into the function and mechanism of the distinct components of the complex. The authors propose a series of mechanistic snapshots based on the structures, presented in Fig 4e.

Response: We agree with the reviewer this is truly seminal work. It is the first structure of a relaxosome, the most complete overview compared to previous work.

Comment 2:

The manuscript describes a large body of work that represents a big step forward in the structural description of the relaxosome of plasmid F and has the potential to inspire many follow up experiments to corroborate and expand the proposed mechanism. is well written and the validation and conclusions are appropriate. I therefore think that the scientific contribution of the paper will be of great value in the field of conjugation of plasmid F.

Response: We agree with the reviewer. One important aspect in the reviewer’s comment is his/her remark on the size of our contribution (“a large body of work”). The paper is already extremely rich data-wise and adding more to it would make the paper unwieldy. Reviewers 1 and 2 do not recommend any additional experiment. Reviewer 3 does not request any additional experiment but finds the biology “limited”. He/she however concludes by saying “despite these limitations, which are often inherent in challenging structural biology projects, this work represents a significant advance in our understanding of bacterial conjugation machinery”.

Comment 3:

As a critical note, the paper does not refer back to previous work and therefore lacks context. I think that this is important to do, because many conjugational systems exist,

each with its particularities concerning the organization of the molecular functions over the different components. Without this context, this work therefore seems of limited interest to groups working on systems other than plasmid F. For example, many structures exist of the relaxase DNA binding domain, not just of Tral, but also for the related TrwC from R388 and from relaxases more distantly related. I suggest a comparison is made between these structures and the TE domain of Tral in the ss-27_+8ds+9_+143-R and ss-27_+8ds+9_+143-R structures.

Response: We agree with the referee. In this revised version, we refer back to previous work (see text in red on pages 11-14). We also present a new Supplementary Fig. 9 that aims to compare the Tral relaxase to the TrwC relaxase as requested by the reviewer. We have added text referring to this comparison and also details pertaining to the larger relaxase field. on page 11-12.

Comment 4:

As a second example of possible context, there is the work published in 2017 by the same group on the same helicase, where they suggest dimerization is necessary for a functional helicase and present the structure of a ssDNA bound to FL Tral. That work is cited in the manuscript, and the mechanism is discussed in the introduction, but the integration of the structures published here and of the previous work is not done (I am thinking of Fig S7 of the previous work for example). I think the paper would be enriched by this analysis.

Response: We are a little puzzled by this comment as we use an entire figure (Fig. 5) to compare the new Tral_{TE} mode to the one described in the 2017 work i.e. Tral_{Helicase}. Also, similar to Fig. S7 from the prior work (Ilangovan et al., 2017), Fig. 6e of this manuscript includes a schematic illustrating the recruitment of two relaxases, essentially integrating earlier findings to provide an overview of the mechanism. The step-wise loading of the two relaxases is shown, with step 5 highlighting the loading of the second relaxase as a helicase. Nevertheless, this manuscript presents a more detailed analysis in light of the work illustrated in Fig. S7 of the previous study.

Comment 5

The structural detail on the conformational change in the relaxase to go from the TE to the relaxase mode is in my opinion one of the nice features of this work. This change is induced by binding of the TE domain to ssDNA. Apparently, the TE mode is not compatible with the structure of the full relaxosome? I think the readers would be very interested in a description of the allosteric mechanism that causes the conformational transformation of the VH domain.

Response: We are a little confused by this comment. Does the reviewer refer to the “relaxase” mode as the architecture of the relaxase as observed in the ds170? What is meant by “apparently, the TE mode is not compatible with the structure of the relaxosome”? We’re unable to offer a response to this comment. However, we did go through the text again to make sure everything we describe is crystal clear.

Comment 6:

One of the revelations of the full relaxosome structure is that the relaxase spans two DNA arms on both sides of the IHF binding site, using the TE and the helicase domains respectively. However, the C-terminal domain of many other helicases are variable and many relaxase families lack helicase domains at their C-terminus. Therefore, this structure suggests that relaxosomes in other systems may have a completely different

structure.

Response: We agree with this statement and have added a sentence to reflect our agreement, taking the R388 relaxosome and the RelSt3 relaxase as examples (see pages 11 and 12).

Comment 7:

As I stated above, I think this work will spark many follow up studies as it raises some interesting points on the relaxosome assembly and function. For example, TraM is often included as a relaxosome protein, but the current work suggests that it is much less involved in the relaxosome complex as there are very few interactions with the Tral, TraY, IHF proteins and their DNA loci. Does this result fit previous consensus or is it a new insight? Related to this, how accurate is the TraM C terminal position, considering that the B factors are very high, consistent with the linker connecting N- and C-terminal is very flexible and there are no interactions?

Response: We agree that our study is seminal in that it will spark many follow up studies. Concerning TraM, indeed, TraM in our structure makes very sparse interactions with the other relaxosome proteins. We have now added an entire paragraph on Page 11 to discuss TraM and its role in relation to the relaxosome structure and its position as a link with the secretion apparatus. As to the TraM position in the relaxosome, it was established by rigid body fitting inside a low-resolution density. Confidence concerning this location is greatly enhanced by the fact the known TraM binding site on DNA is situated next to this density.

Minor comment 1:

- Why is the Tral sequence cut up into different chains in the pdb structure? I suggest that the same chain identifier be used for all residues of the Tral

Response: This has been done and the new PDB can be found in the ChimeraX session folder.

Minor comment 2:

- The names used in the paper for the different structure should be included in fig 1c-e

Response: The labels have been added in what is now Fig. 2c-e.

Minor comment 3:

- Lines 231 to 234, "The AH and CTD domain ... ie is semi flexible" should be revised.

Response: Indeed, the sentence has problems. We have split it in 2 (see page 13)

Minor comment 4:

- The use of TralTE as mode and Tral TE as domain can be confusing. Eg, in line 239, specify that this refers to the domain

Response: Indeed, on this occasion, it is not clear. We have added "domain".

Reviewer #2 (Remarks to the Author)

Comment 1:

Conjugation is an important research topic due to its role in the spread of antibiotic resistance. The process involves a type 4 secretion system (T4SS) that spans the double membrane, a conjugative pilus, and a protein complex called the relaxosome, which binds to and processes the plasmid before it is transferred to a new host cell.

While recent research has provided insights into the structure of the T4SS and pilus, the structure of the relaxosome has not been resolved. The present study by Williams et al fills this gap by reporting the cryo-EM structure of the fully assembled relaxosome encoded by the paradigm F plasmid. The authors report two distinct complex states, which they suggest represent different functional steps in plasmid DNA processing. These findings are intrinsically novel and contribute valuable new information to the field, summarized in a comprehensive model (Fig. 4) for conjugation steps.

Response: We agree the study fills a crucial gap in our knowledge of T4SS and conjugative transfer. Also, we agree that the findings are intrinsically novel.

Comment 2:

However, the manuscript would benefit from a more accessible presentation, a deeper discussion of mechanistic insights and their biological implications, as it currently lacks a clear narrative to help guide the reader and highlight the significance of these new structures.

Response: In this revised version, we have attempted to address this comment by highlighting more of the previous knowledge (in red, pages 11-14). We note that the reviewer in his/her next comment states that the paper follows a rigorous and logical experimental approach, which, in our view, constitute (and should constitute) the backbone of the narrative.

Comment 3:

While the experimental approach is logic and the structural insights well-articulated, more discussion on how these findings differ or align with existing knowledge in relaxosome assembly and binding to the oriT would strengthen the manuscript's impact. The manuscript would benefit from a comprehensive discussion on whether the structural findings are likely unique to the F plasmid or could be (even partially) generalizable to other plasmids.

Response: This is an interesting point. We have now included a discussion regarding whether our findings are unique to F or could be generalizable to other plasmid systems. This is now discussed in numerous additions in the text shown in red (pages 11, 12 and 13).

Comment 4:

The study provides a detailed structural breakdown of the relaxosome's hubs, but the functional significance of these hubs could be clarified and expanded. The manuscript emphasizes the flexibility of certain relaxosome components, particularly Tral helicase and trans-esterase domains, but this point could be explored further. The manuscript shows that mutations in specific residues within the protein-DNA and protein-protein interaction hubs either disrupt or enhance conjugation efficiency. Providing a more detailed explanation of why certain residues are particularly critical (e.g., conserved catalytic sites in Tral) would solidify the structure-function relationship. The role of DNA bending and unwinding (e.g., U-turns induced by IHF and Tral) in relaxosome assembly is described. How critical are these bends to the precise positioning of the "nic" site and subsequent DNA unwinding?

Response: this comment contains several subsections. Here is our response:

4.1. The manuscript emphasizes the flexibility of certain relaxosome components, particularly Tral helicase and trans-esterase domains, but this point could be explored further.

Response to 4.1: We respectfully disagree with this comment. We make a great deal out of the flexibility of the structure. So we are not sure how it could be emphasized more than it is presently.

4.2. The role of DNA bending and unwinding (e.g., U-turns induced by IHF and Tral) in relaxosome assembly is described. How critical are these bends to the precise positioning of the "nic" site and subsequent DNA unwinding?

Response to 4.2: The answer is that we have not attempted to modify the bending of DNA, something that would be very difficult, possibly impossible, to do. Therefore we don't have an answer to this question. We can only speculate that bending is crucial for positioning the various components of the relaxosome in a way that maximalise function (pages 6-7). We have added a sentence on page 6 stating the following: "Bends induced by IHF and accessory proteins may position the *nic* site for subsequent cleavage by Tral".

Comment 5:

The methodology is thorough but highly technical. A summarized explanation of cryo-EM's relevance and why it was chosen over other structural techniques would improve clarity. Figures are well-integrated and detailed. However, descriptions could be simplified in figure legends to make them more self-contained for readers less familiar with the methods.

Response: We now explained why cryo-EM is relevant and why it was chosen over other structural techniques. It now reads on page 6: "Cryo-EM is the most suitable structural biology method to use in this case as the structure is modular and cannot be crystallised". We are somewhat puzzled by the idea of simplifying the figures by making them more self-contained. It seems to us that making them self-contained will require a lot more details than presently described, therefore resulting in figures difficult to read and also many repetitions, something that should probably be avoided. So we have left the figures legends as they were in the previous version.

Reviewer #3 (Remarks to the Author)

Comment 1:

This manuscript presents the first high-resolution structural analysis of the bacterial relaxosome, a nucleoprotein complex involved in bacterial conjugation. Using cryo-electron microscopy and modeling, the authors reveal detailed structures of the fully assembled F plasmid relaxosome in its pre-activation state, providing new structural insights into bacterial DNA transfer machinery. Given that plasmids are major carriers of antibiotic resistance genes, this work has direct implications for understanding resistance spread.

The relaxosome consists of three plasmid-encoded proteins (Tral, TraY, and TraM) and one host-encoded protein (IHF) assembled on a specific DNA sequence called *oriT* (origin of transfer). The authors determined the structure of the relaxosome in its pre-activation state. Using different DNA constructs, they investigated how the complex recognizes and processes its DNA substrate. The structure reveals an intricate organization where the DNA forms an asymmetric U-shaped hairpin, with proteins arranged in four distinct "hub" regions that combine protein-DNA and protein-protein interactions. The central protein Tral is observed in a 'TE mode' (trans-esterase mode) conformation, which differs significantly from its previously known 'helicase mode' structure.

The structural analysis reveals extensive interaction networks between the protein components and DNA. The authors mapped multiple protein-protein interfaces and protein-DNA contacts, providing atomic-level detail of how the complex is assembled. A particularly notable feature is the DNA architecture, which includes two dramatic U-turns: one in the double-stranded DNA region induced by IHF binding, and another in the single-stranded region upon Tral TE binding.

Mutational analysis of Tral interfaces was performed, showing that disrupting certain protein-protein interactions affects conjugation rates. Based on their structural observations, the authors propose a mechanistic model for bacterial conjugation, though this model would require additional functional validation.

This work provides the first detailed structural framework for understanding relaxosome assembly. While the functional implications of these structures remain to be fully explored, they offer a foundation for future studies of bacterial conjugation mechanisms and potentially inform strategies to combat antibiotic resistance spread.

Response: As also stated in comments by Reviewers 1 and 2, our investigation of the relaxosome is indeed the first and also seminal as they both agree that it offers foundation for future studies. However, Reviewer 3 states that the study has several important limitations to consider. Nevertheless, Reviewer 3 does not request any additional experiments. He/she notes: "This work provides the first detailed structural framework for understanding relaxosome assembly. While the functional implications of these structures remain to be fully explored, they offer a foundation for future studies of bacterial conjugation mechanisms and potentially inform strategies to combat antibiotic resistance spread".

Comment 2:

The functional data in the paper is quite limited. Conjugation frequency assays for Tral mutants (Fig. 4a-d), measuring transfer rates when key interface residues are mutated. While this provides some functional validation of the structural observations, these are relatively basic assays focused only on Tral mutations. DNase I footprinting experiments (Extended Data Fig. 2) to map protein binding regions on oriT DNA. However, this is more of a structural validation rather than true functional analysis. The manuscript lacks several types of functional studies that would strengthen its conclusions, such as direct biochemical assays showing the proposed DNA bubble formation and its role in activation. While these additional experiments would have strengthened the mechanistic model, they are probably beyond the scope of this structural study.

Response: We agree with the reviewer's statement. As the reviewer states earlier, the relaxosome structure we present here is the first of any relaxosome, and will inevitably serve as a springboard for future investigations, including the more detailed biochemical ones that the reviewer describes in his comment. As the reviewer him/herself point out: these are beyond the scope of the structural study presented here. Therefore, the reviewer does not request any additional experiment and, as a result, we do not present any additional biochemical and genetic results. However, we present a new structure that completes our investigation of bubble requirement. Indeed, we have now added a new DNA in this part of our investigation: a fully double-stranded DNA, ds-2_+113, that encompasses the TE-binding sequence showing that DNA melting at the TE-binding site does not necessarily requires a ssDNA overhang. We consider the use of this DNA as an essential complement to the experiment that makes use of the ds-67_+113(poly-dT15-17_-3) bubble DNA. The text has been modified

accordingly to describe the rationale, experimental design, and results of this new experiment (highlighted in green, pages 14-15).

Comment 3:

While the structural work is impressive, the manuscript misses opportunities to integrate these new findings with the extensive existing literature. Previous biochemical and genetic studies had identified key residues in Tral, characterized IHF's role in DNA bending, and mapped essential regions of oriT - yet there is limited discussion of how this structure explains or challenges these findings. Similarly, while the authors cite structures of relaxases from Gram-positive bacteria and mention the R1 plasmid system, they don't provide detailed structural comparisons that could have provided broader mechanistic insights into bacterial conjugation systems.

Response: We have now added a more substantial discussion to describe how our findings relates to the results published in previous studies (in red, pages 11-14). See response to similar comments by Reviewers 1 and 2.

Comment 4:

The complex was reconstituted *in vitro* from purified components rather than isolated from cells and chemically cross-linked with glutaraldehyde, which could potentially trap non-physiological conformations. The authors used a catalytically inactive mutant (Tral Y16F) which might affect the native structure. The bias or limitations of using a mutant, *in vitro* reconstitution and glutaraldehyde crosslinking was not explicitly discussed in the paper.

Response: We respectfully disagree with the reviewer. A myriad of biochemical studies have demonstrated the relevance of *in vitro* reconstitution work from purified components in providing mechanistic insights on biological processes involving macromolecular machines. Thus, *in vitro* work does not constitute a limitation, but a means to gain precious mechanistic information, sometimes the only one. We have added on page 16: "A myriad of biochemical studies have demonstrated the relevance of *in vitro* reconstitution work from purified components in providing mechanistic insights on biological processes involving macromolecular machines". As to using a catalytically inactive mutant and crosslinking, we have now added the following on page 23: "A nickase inactive version of Tral harbouring Y16F mutation (Tral_{Y16F}) was used for all cryoEM complex formations⁵⁵ in order to prevent the forward reaction that might destabilised the complex" and page 24; "The peak fractions containing the relaxosome were pooled (1 ml total volume) to which Glutaraldehyde (Grade I, 25% in H₂O, Sigma) was added to a final concentration of 0.5% to increase complex stability under cryo conditions." We have also added in the legend to Fig. 1a: "We do not believe that crosslinking introduces bias in our case, since our structure recapitulates prior biochemical and structural knowledge on individual proteins".

Comment 5:

Another limitation of this study is that it primarily captures a single "quiescent" pre-activation state of the relaxosome, leaving critical questions about the complex's dynamic behavior unanswered. The absence of structures representing the active state, particularly after cell-cell contact triggers conjugation, creates a significant gap in our mechanistic understanding. Furthermore, several functionally important domains, including the Active Helicase (AH) and C-Terminal Domain (CTD), remain

partially disordered in the structures, limiting our insight into their roles during DNA processing and transfer. This could be discussed.

Response: We agree that our study, although providing the first view of a relaxosome, only presents the structures of a few of the many states that the relaxosome, and more particularly the relaxase must adopt before conjugation is completed. We have now added the following text on page 16: “Here, we present not only the first structure of a relaxosome but also describe various DNA-bound states of this important complex. The structural characterisation of many more conformational states will be required to gain a full understanding of relaxosome function. Nevertheless, from the insights obtained here, a molecular-level mechanistic model for relaxosome recruitment and activation emerges (Fig. 6e)”.

Comment 6:

The study relies heavily on computational approaches and modeling to generate its structural conclusions. The authors used AlphaFold2-predicted models for fitting several components into their cryo-EM maps and employed extensive computational modeling to interpret poorly resolved regions. Notably absent are any quality metrics for these AlphaFold2 models, such as pLDDT scores, PAE plots, making it difficult to assess the reliability of the computational models used for structural interpretation.

Response: We now provide in Supplementary Fig. 5a-d all the AlphaFold pLDDT scores mapped onto the structures we used as initial model before refitting into density.

Comment 7:

Despite these limitations, which are often inherent in challenging structural biology projects, this work represents a significant advance in our understanding of bacterial conjugation machinery.

Response: We are pleased that the reviewer considers our structure of the relaxosome as a significant advance that will stimulate research in the field.

In summary, we have addressed all comments and suggestions by all reviewers. We thank them for their very positive contribution to the improvement of our manuscript. We believe the manuscript is now ready for publication in Nature Communications.

Yours, sincerely

Gabriel Waksman

Reviewer #1 (Remarks to the Author):

Reviewer's general comment:

The authors made an effort to address on my comments, which I appreciate. I apologize if my previous comments were not clear. They were made because I find the conformational rearrangement of Tral in the TE and helicase mode remarkable and believe that this deserves more attention than that given in the current manuscript.

The VH domain relocates from the exterior of the protein in the Tral(helicase) mode, to the boundary between the TE and AH domains in the Tral(TE) mode. In addition, the VH domain itself undergoes a large rearrangement (Fig 5b). Page 13 line 284-285 state that this part of the protein does not form interactions with nearby molecules and page 13, lines 282-283 state that the AH and CTD domains are largely unstructured. It is therefore not clear what brings about this change and the manuscript could clarify better on what basis the VH and AH were built like this. In particular, I hope the authors can address the following points:

Comment 1

Fig 5A and Fig S1 only shows a schematic representation of the Tral(TE) and the Tral(helicase) structures, but there are no density maps of the Tral(TE) mode, or superpositions of the domains of the two structures in the current manuscript. I suggest to include in a supplementary figure i) density maps of the regions mapping to the individual domains of Tral(TE), ii) superpositions of the TE and AH models in TE and helicase modes, where possible, to map the changes in these domains that allow the relocation of the VH domain.

Response to comment 1:

We respectfully disagree: Fig 5A and Fig S1 DO NOT show schematic diagrams but REAL structures in surface representation. As to the density for the structure of the Tral_{TE} mode, it is already shown in Figures 3a and S4h. So we do not feel that a repeat of those figures will be helpful. As to the requested superposition, this is shown in a much better way in Figure 5a where we show the Tral_{TE} mode and the $\text{Tral}_{\text{Helicase}}$ one in two different panels BUT with the same orientation for the TE domain of both structures. Thus, there is really no need for an additional supplementary figures since all reviewer's request can be found in the existing figures and supplementary figures.

Comment 2:

Given the resemblance between the structures of the VH and AH (Langovan et al. 2017 page 714), I wonder if the density around the VH and AH is sufficiently clear to distinguish between the two, can the authors comment on that? Perhaps show clear evidence of density in regions that help distinguish between the VH and AH (see also my previous point)?

Response to comment 2:

It is a legitimate concern and we now provide in a new panel in Supplementary Figure 5 (new panel e) a superposition of the $2A+2B/2B$ -like subdomains of both the VH and AH domains together with the density we have obtained for this region. This new figure panel clearly shows that $\text{AH}_{2A+2B/2B\text{-like}}$ contains additional secondary structures, the density of which is clearly absent. In contrast, all secondary structures for $\text{VH}_{2A+2B/2B\text{-like}}$ are present in the density. We are therefore very confident that the domain assignment is correct. In addition to providing a new panel in Supplementary Fig. 5, we have amended the text the following way: 1- Supplementary Fig. 5efg are now

Supplementary Fig. 5fgh; 2- we have added in the Methods section (in green) on page 30 “It should be noted that although the $VH_{2A+2B/2Blike}$ and $AH_{2A+2B/2Blike}$ superpose well, there are loops and secondary structure elements essential for ATP and ssDNA binding present in the latter, but absent in the $VH_{2A+2B/2Blike}$ sub-domain (Supplementary Fig. 5e).”

Comment 3:

I would suggest an analysis of the possible allosteric pathways for the transition between the Tral(TE) and Tral(helicase). What conformational changes in the protein allow for the relocation of the VH? How are the interactions surfaces between the TE domain and the AH domain with the VH domain in TE mode? Have similar conformational rearrangements been described for helicase domains relate to VH?

Response to comment 3:

This is a fascinating issue. However, describing the allosteric pathway from the Tral_{TE} mode to Tral_{Helicase} mode would require an entire new set of investigations which are best left for another paper. Also, some of the regions in the Tral_{TE} mode are flexible, rendering the analysis of the allosteric pathway even more difficult, necessitating the use of molecular dynamics in order to get a handle on the conformational change continuum required to go from one form to the other for these regions.

Comment 4:

Concerning the added Suppl Fig 9, showing the superposition of R388 TrwC AH with the Tral VH of the Tral in TE mode: is the TrwC AH domain in open (aka TE mode) or closed form (aka helicase mode)?

Response to comment 4:

The R388 TrwC structure shown in Supplementary Fig. 9a is an alphafold model, which is modelled on Tral_{Helicase} i.e. closed conformation. As to the superposition in Supplementary Fig. 9e, we modelled TrwC in an open form based of the structure of the the Tral_{TE} mode presented here. We have amended the legend to Supplementary Fig. 9a and e accordingly.

Minor comments:

Minor comment 1:

The 2017 paper describes an open and a close conformation of the relaxase. I suggest the authors explicitly link the terms "open" and "closed" state, used in the 2017 cell paper, to the different states reported in this manuscript.

Response to minor comment 1:

We have added a new paragraph to the section, now referring to the “open” and “closed” form of Tral. It says (new text in green on pages 12 and 13): “In a previous study¹⁷, we characterised two forms for Tral (Supplementary Fig. 1c,d). The first form was observed when Tral was bound to the ssDNA sequence 5’ of the *nic* site (*tral*_{TE} in Fig. 1b). It is sensitive to mild-proteolysis i.e. it adopts a flexible and “open” structure. In contrast, the second form is protease-resistant, i.e. rigid and “closed”, and is observed only when Tral is bound to the ssDNA sequence 3’ of the *nic* site (*tral*_{helicase} in Fig. 1b). The structure of Tral bound to *tral*_{helicase} was solved¹⁷: in this structure, *tral*_{helicase} is bound through the helicase domains and this closed form of Tral was designated the Tral_{Helicase} form of Tral.

With the ss-27_+8ds+9_+143-R relaxosome structure, we capture a state of Tral where the protein engages with its TE-binding site i.e. the sequence 5’ of the *nic* site. It is also

very flexible since the AH and CTD domain of Tral remain largely semi-unstructured. It therefore captures the open form characterised previously¹⁷. We name this form of Tral "Tral_{TE}". In this conformational state, Tral has undergone a dramatic re-arrangement of its domain and subdomain structure compared to its helicase mode (Fig. 5a, b): i- while the AH domain was proximal to the TE domain in Tral_{Helicase}, in Tral_{TE}, it is the VH domain that is now proximal to that domain (Fig. 5a); ii- In Tral_{TE}, the VH domain itself has undergone a large re-arrangement of its sub-domains, with the NTD and the 2A+2B/2B-like module pivoting 47° and 95° degrees each relative to the 1A sub-domain (Fig. 5b). Near atomic resolution was not achieved for the AH and CTD domains, nonetheless we were able to locate these domains within the relaxosome. Apparently, this region makes no specific and stabilising contacts with proteins or DNA nearby."

Minor comment 2:

page 16, line 368: change "the TE of Tral_{TE}" to "the TE domain of Tral_{TE}"

Response to minor comment 2:

Done

Minor comment 3:

page 9, line 198: change "Tral TE is observed" to "Tral TE domain is observed"

Response to minor comment 3:

Done

Minor comment 4:

page 10, line 212 "from unbound to TE-bound": clarify that the TE domain is meant here

Done

page 10, line 216: change "imposed by its interaction with TE" to "imposed by its interaction with the TE domain"

Response to minor comment 4:

Done

Minor comment 5:

page 11, line 245: change "with IHF and Tral TE" to "with IHF and the Tral TE domain"

Response to minor comment 5:

Done